# Less intensive antileukemic therapies (monotherapy and/or combination) for older adults with acute myeloid leukemia who are not candidates for intensive antileukemic therapy: A systematic review and meta-analysis

Luis Enrique Colunga-Lozano[1¤]*, Fernando Kenji Nampo[2], Arnav Agarwal[3], Pinkal Desai[4], Mark Litzow[5], Mikkael A. Sekeres[6], Gordon H. Guyatt[1], Romina Brignardello-Petersen[1]

1 Department of Health Research Methods, Evidence and Impact, McMaster University, Hamilton, Ontario, Canada, 2 Department of Latin-American Institute of Life and Nature science, University of Latin-American Integration, Foz Do Iguaçu, Parana, Brazil, 3 Department of Medicine, Toronto University, Toronto, Ontario, Canada, 4 Division of Hematology and Medical Oncology, Weill Cornell Medical Center, New York, New York, United States of America, 5 Division of Hematology, Mayo clinic, Rochester, Minnesota, United states of America, 6 Division of Hematology, Sylvester Comprehensive Cancer Center, University of Miami, Miami, Florida, United States of America

¤ Current address: Department of Clinical Medicine, Universidad de Guadalajara, Guadalajara, Jalisco, México
* dr.colunga.lozano@gmail.com

## Abstract

### Introduction

Elderly patients with acute myeloid leukemia not eligible for intensive antileukemic therapy are treated with less intensive therapies, uncertainty remains regarding their relative merits.

### Objectives

To compare the effectiveness and safety of less intensive antileukemic therapies for older adults with newly diagnosed AML not candidates for intensive therapies.

### Methods

We included randomized controlled trials (RCTs) and non-randomized studies (NRS) comparing less intensive therapies in adults over 55 years with newly diagnosed AML. We searched MEDLINE and EMBASE from inception to August 2021. We assessed risk of bias of RCTs with a modified Cochrane Risk of Bias tool, and NRS with the Non-Randomized Studies of Interventions tool (ROBINS-I). We calculated pooled hazard ratios (HRs), risk ratios (RRs), mean differences (MD) and their 95% confidence intervals (CIs) using a random-effects pairwise meta-analyses and assessed the certainty of evidence using the Grading of Recommendations Assessment, Development, and Evaluation (GRADE) approach.

**Data Availability Statement:** All relevant data are within the manuscript and its Supporting Information files.

**Funding:** The American Society of Hematology provided funding for the development of the clinical practice guideline related to this topic (American Society of Hematology 2020 guidelines for treating newly diagnosed acute myeloid leukemia in older adults), however, no funding was used during the development of this study. Information added as "Related Manuscript".

**Competing interests:** The authors have declared that no competing interests exist related to this work. This does not alter our adherence to PLOS ONE policies on sharing data and materials.

## Results

We included 27 studies (17 RCTs, 10 NRS; n = 5,698), which reported 9 comparisons. Patients were treated with azacitidine, decitabine, and low-dose cytarabine (LDAC), as monotherapies or in combination with other agents. Moderate certainty of evidence suggests no convincing difference in overall survival of patients who receive azacitidine monotherapy compared to LDAC monotherapy (HR 0.69; 95% CI, 0.31–1.53), fewer febrile neutropenia events occurred between azacitidine monotherapy to azacitidine combination (RR 0.45; 95% CI, 0.31–0.65), and, fewer neutropenia events occurred between LDAC monotherapy to decitabine monotherapy (RR 0.62; 95% CI 0.44–0.86). All other comparisons and outcomes had low or very low certainty of evidence.

## Conclusion

There is no convincing superiority in OS when comparing less intensive therapies. Azacitidine monotherapy is likely to have fewer adverse events than azacitidine combination (febrile neutropenia), and LDAC monotherapy is likely to have fewer adverse events than decitabine monotherapy (neutropenia).

## Introduction

Acute myeloid leukemia (AML) is a heterogeneous hematopoietic stem cell cancer with incomplete maturation of blood cells and a reduced production of normal hematopoietic elements [1]. AML is more common in older adults with a median age at diagnosis of 67 years old; one-third of cases occur in patients older than 75 years [2].

Overall survival (OS) is strongly linked to clinical and biologic characteristics; age, performance status (PS), karyotype, mutational status and response to induction therapy [3]. For example, younger patients (2 to 30 years) have a much better 5-year OS than older patients (65 to >85 years) (57% to 42%, compared to 6.8% to 1.2%) [4,5].

Some older patients diagnosed with AML are not eligible for intensive treatment, limiting their therapeutic options [6]. Less intensive therapy with hypomethylating agents or low-dose cytarabine, as examples, has been used to treat older AML patients who are not candidates for intensive therapy [7].

In their 2020 guidelines, the American Society of Hematology (ASH) provided recommendations for the treatment of older adults with newly diagnosed AML who are considered appropriate for antileukemic therapy, but not intensive antileukemic therapy [8]. When choosing between monotherapies, the guideline panel conditionally recommended the use of either hypomethylating-agents (azacitidine or decitabine) or low-dose cytarabine and, when choosing between monotherapies or combinations, the guideline panel conditionally recommend using monotherapy [8].

To inform the recommendations provided by the ASH 2020 guideline for Treating Newly Diagnosed Acute Myeloid Leukemia in Older Adults [8]. We conducted a systematic review to compared the comparative effectiveness and safety of low-intensity antileukemic therapies (monotherapy and/or combination) in older adults with newly diagnosed AML who are not candidates for intensive therapy.

## Methods

### Protocol and registration

This systematic review was not registered on PROSPERO or other registries. This systematic review was performed with ASH guideline methodology [9] and informed the development of recommendations regarding the treatment of AML in elderly patients from the ASH 2020 Guidelines for treating newly diagnosed acute myeloid Leukemia in Older adults [8]. The eligibility criteria for studies to include were pre-established by the panel when formulating the recommendation questions. We conducted the study in accordance with the Cochrane handbook [10] and report the results according to the Preferred Reporting Items for Systematic Reviews and Meta-Analyses guidelines [11] (S1 Checklist).

### Eligibility criteria

We included randomized clinical trials (RCTs) and comparative non-randomized studies (NRS) of adults 55 years or older, with newly-diagnosed AML published in any language comparing the following less intensive therapies against each other, either as a monotherapy or in combination with any secondary agent: gemtuzumab ozogamicin, low dose cytarabine (LDCA), azacitidine (AZA) and decitabine (DEC). Outcomes of interest were mortality, quality of life, functional status, recurrence, morphologic complete remission, severe toxicity (CTC adverse effects grade 3 or higher), or burden on caregivers, measured in any way. We excluded studies that enrolled patient with acute promyelocytic leukemia, or myeloid proliferations related to Down syndrome and those in which researchers combined any of the interventions of interest with any agent considered a component of intensive antileukemic therapy regimens. Detailed description of the eligibility criteria—*type of studies*, *participants*, *interventions and outcomes*- is reported in S1 Appendix.

### Information sources and search

We searched MEDLINE and EMBASE from inception to August 2021 without restrictions on language of publication. For informing the ASH recommendations, we searched for studies published through July 2019.

We conducted an umbrella search that encompassed all the questions addressed in the guideline [8]. The supporting information file describes the search strategies items (S2 Appendix). We checked the reference lists of reviewed studies and contacted clinical experts for additional references.

### Study selection and data collection process

Pairs of reviewers screened titles and abstracts obtained through the electronic searches and identified those potentially eligible. We then grouped studies according to the question they addressed and conducted full text screening specifically for our question. Four reviewers, independently working in pairs (BPR, NKF, AA, LECL) made eligibility decisions. If reviewers could not resolve disagreement through discussion, a third reviewer adjudicated (RBP).

Pairs of reviewers independently abstracted data on a standardized form. We extracted the following information: type of study, recruitment time-frame, follow-up (months), sample size, participant characteristics, as age (years), gender, cytogenetics (intermediate or poor), performance status (ECOG or WHO classification), white cell count, AML diagnosis criteria, trial location, source of funding, trial registry interventions (main agent, dose and second agent for combination therapy groups), comparisons (main agent, dose and second agent for

combination therapy groups), and outcomes (mortality, quality of life, functional status, recurrence, morphologic complete remission, severe toxicity (CTC adverse effects grade 3 or higher), or burden on caregivers, at any time point. If reviewers could not resolve disagreement through discussion, a third reviewer adjudicated (RBP).

## Risk of bias in individual studies

Pairs of reviewers (BPR, NKF, AA, LECL), independently assessed risk of bias for each randomized controlled trial using a modified version of the Cochrane risk of bias tool for randomized trials [12] and, for nonrandomized studies, the Risk of Bias Assessment Tool for Non-Randomized Studies of Interventions ROBINS-I tool [13].

## Data analysis

We calculated the relative effect of less intensive therapies using hazard ratios (HR) for time to event data, relative risk (RR) for dichotomous outcomes, and mean difference for continuous outcomes, with their 95% confidence intervals (CIs). We used random-effects models with the DerSimonian-Laird estimate of heterogeneity to pool data across studies reporting the same comparison and outcome [10]. We used forest plots to display comparisons with two or more pooled studies. We carried out all statistical analyses using Review Manager 5.3 [14]. We planned to conduct a network meta-analysis to compare all interventions against each other, but there was no sufficient data to conduct such analysis (data not shown). We analyzed data from RCTs and NRSs separately.

## Dealing with missing data

When details about study design or descriptive statistics for outcomes were not presented in original publications, we did not impute data but rather contacted authors for additional information.

## Assessment of the certainty of evidence by outcome

We used the Grading of Recommendation, Assessment, Development, and Evaluation (GRADE) methodology to rate the certainty of evidence (also known as quality of evidence) for each outcome as high, moderate, low, or very low [15]. The assessment included judgments addressing risk of bias, imprecision, inconsistency, indirectness, and publication bias [15]. In addition, we assessed the magnitude of the effect, the presence of dose-response relationships, and residual confounding when rating the certainty of evidence from NRS [16]. We estimated absolute effect measures to facilitate the decision-making process [17]. Using absolute effects that we calculated based on the baseline risk of the comparator arms in the included studies, we rated the certainty that there was any benefit or any harm using a minimally contextualized approached [18]. We rated down due to imprecision if the confidence intervals crossed the null effect, and if the effect estimate was obtained from a small number of participants or events [19]. We assessed inconsistency between studies by visual inspection of forest plots, in particular the extent of overlap of confidence intervals (CI), the Q statistic (with a p value $\leq 0.1$ as a suggestion of important statistical heterogeneity), and the $I^2$ value [20]. We planned, if ten or more studies were available for a particular outcome, to create a funnel plot to assess publication bias by visual inspection [21]. Because we had multiple comparisons, we created Summary of Findings Tables for each comparison [22] and outcome using GRADEpro GDT (www.gradepro.org) [23].

### Subgroup and sensitivity analysis

We pooled and reported results from RCTs and NRS separately. We planned to conduct sensitivity analyses to explore the impact of the risk of bias in the effect estimates. We performed a subgroup analysis to explore the impact of the secondary agent (when comparing a combination therapy group) in the effect estimates, when there were sufficient studies. The number of studies per comparisons did not allow us to explore any subgroup analysis based on patients' characteristics (e.g., gender)

## Results

### Search results

After the removal of duplicates, we identified 12,376 studies of which 149 proved to be potentially relevant based on title an abstract screening. After full text review, we included 27 studies (Fig 1). From the included studies, 21 were included after the first search and informed the development of the recommendations [24–43], 6 studies were included after the guideline recommendation [44–49] We did not find any ongoing studies.

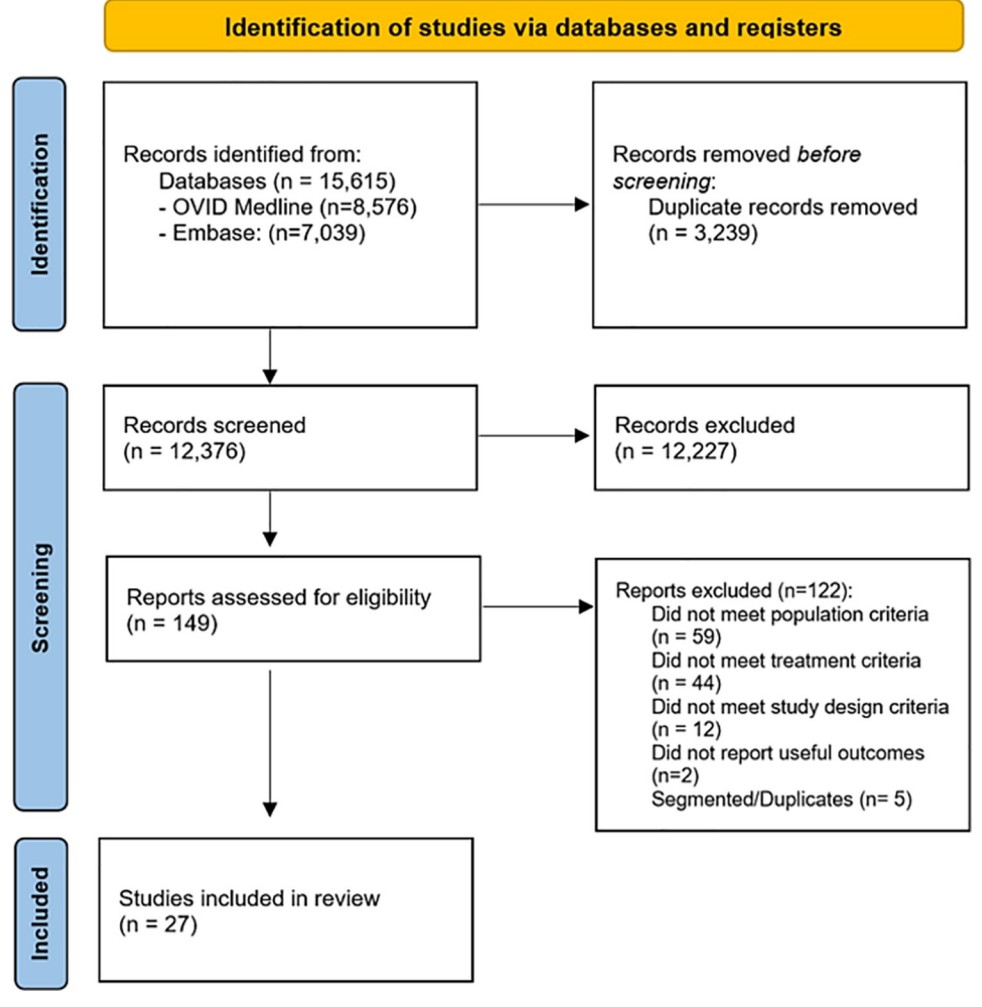

**Fig 1. Eligibility assessment PRISMA flow diagram.**

## Study characteristics

We included 27 studies: 17 RCTs (3,902 patients) [24–35,37,46–49] and 10 NRS (1,796 patients) [36,38–42,44,45,50,51] published between 2007 to 2020. Table 1 and S3 Appendix summarize the study characteristics. 12 studies were single center (Four RCTs [26,27,29,33], one prospective NRS [36] and seven retrospective NRS [38–40,42,43,45,51]) and 15 were multi-center studies (13 RCTs [24,25,28,30–32,34,35,37,46–49] and 2 prospective NRS [41,44]). The trials' geographical location is reported in S3 Appendix. Participants median age ranged from 67 years to 76 years, female participation ranged from 20% to 57.2%, and follow-up ranged from 3.3 months to 54 months.

The criteria to diagnose AML varied across studies; 14 used the WHO-AML criteria [8,30–34,37,41,44,46,47,49,50], six did not provided information [25–27,39,40,45], five studies used bone marrow blast percentage description [28,35,36,38,51], one used the National Comprehensive Cancer Network 2009 criteria [24], and one used immunophenotype confirmation [42].

We identified 9 comparisons from the 27 included studies: two RCTs (433 patients) compared azacitidine monotherapy against low-dose cytarabine monotherapy [24,25]; four RTCs (921 patients) compared azacitidine monotherapy against azacitidine in combination with a second agent (venetoclax [48], entinostat [34] and vorinostat [32,33]); three NRS (648 patients) compared azacitidine monotherapy against decitabine monotherapy [39,40,50]; three RCTs (685 participants) compared decitabine monotherapy against decitabine in combination with a second agent (bortezomib [37], valproate and/or retinoic acid [46] and talacotuzumab [47]); two NRS (190 patients) compared decitabine in combination with a second agent (venetoclax) against azacitidine in combination with a second agent (venetoclax) [41,44]; seven RCTs (1406 patients) and one NRS (28 patients) compared low-dose cytarabine monotherapy against low-dose cytarabine in combination with a second agent (ATRA [51], arsenic trioxide [26], gemtuzumab ozogamicin [27], lintuzumab [29], volasertib [30], vosaroxin [28], glasdegib [31], and venetoclax [49]); one RCT (457 patients) and one NRS (30 patients) compared low-dose cytarabine monotherapy against decitabine monotherapy [35,36]; one NRS (406 patients) compared low-dose cytarabine in combination with a second agent (not specified) against hypomethylating agents [38]; and, two NRS (485 patients) compared low-dose cytarabine monotherapy against hypomethylating agents [42,45]. Meta-analyses were done reporting each comparison and without mixing the study designs.

## Risk of bias of the included studies

We provide a detailed description of the risk of bias assessment per study and domain in S4 Appendix. All NRS had serious risk of bias due to confounding because patient baseline characteristics were different between the treatment groups [36,38–42,44,45,50,51]; two of the 10 studies had bias in the selection of participants into the study (serious (36) and moderate [51]); three of the studies had moderate risk of bias in the classification of the interventions, [38,42,45]; and seven of the studies had bias due to deviations from the intended interventions (serious [42] and moderate[38,39,41,43–45]). None of the studies had risk of bias due to missing data, outcomes measurements and selective reporting (S4 Appendix). All RCTs had low or probably low risk of bias in the sequence generation domain [24–35,37,46–49]; three of the 17 studies had high risk of bias in the allocation concealment domain [26–28]; all the studies had low or probably low risk of bias in the blinding domains (performance and outcome measurement), missing data and selective reporting (S4 Appendix).

**Table 1. Study characteristics.**

| Author, year | Time frame | Overall Age (y) Median, (Range) | Sample Size | Follow-up, Months (median) | Female Gender n/ N (%) | Main therapy | 2nd agent | Comparison therapy | 2nd agent | AML diagnosis |
|---|---|---|---|---|---|---|---|---|---|---|
| **Randomized controlled trials** | | | | | | | | | | |
| Wei, 2020 [49] | 2017–2018 | 76 (41–88) | 211 | 12 | 94/211 (44.5) | LDAC | Venetoclax | LDAC | NA | WHO Classification |
| DiNardo, 2020 [48] | 2017–2019 | 76 (49–91) | 433 | 20.5 | 172/431 (39.9) | Azacitidine | Venetoclax | Azacitidine | NA | WHO Classification |
| Montesinos, 2020 [47] | 2015–2017 | 75 (65–92) | 316 | 25.1 | 145/316 (45.8) | Decitabine | NA | Decitabine | Talacotuzumab | WHO Classification |
| Lubbert, 2020 [46] | 2011–2015 | 76 (61–91) | 204 | 25.1 | 72/200 (36) | Decitabine | NA | Decitabine | Valproatro, ATRA, or both | WHO Classification |
| Cortes, 2019 [31] | 2014–2017 | G1: 77 (63–92) G2: 75 (58–83) | 132 | 20 | 37/132 (28) | LDAC | Glasdegib | LDAC | NA | WHO Classification |
| Roboz, 2018 [37] | 2011–2013 | 72.4 (60.5–92.3) | 165 | 30 | 50/163 (30.7) | Decitabine | NA | Decitabine | Bortezomib | WHO Classification |
| Montalban bravo, 2017 [33] | 2009–2010 | 70 (30–90) | 79 | 7.4 | 9/30 (30) | Azacitidine | NA | Azacitidine | Vorinostat | WHO Classification |
| Craddock, 2017 [32] | 2012–2015 | Not reported | 260 | 10 | 103/259 (40) | Azacitidine | NA | Azacitidine | Vorinostat | WHO Classification |
| Dennis, 2015 [28] | 2012–2013 | 75 (60–91) | 104 | 17 | 35/104 (33.6) | LDAC | NA | LDAC | Vosaroxin | Bone marrow blast |
| Dohner, 2014 [30] | 2010–2011 | G1: 76 (57–86) G2: 75 (65–87) | 87 | 28.2 | 39/87 (44.8) | LDAC | NA | LDCA | Volasertib | WHO Classification |
| Dombret, 2014 [24] | 2010–2014 | 75 (64–91) | 399 | 24.4 | 166/399 (41.6) | Azacitidine | NA | LDAC | NA | NCCN 2009 criteria |
| Prebet, 2014 [34] | 2006–2010 | 72 (25–87) | 149 | 30 | 47/149 (31.5) | Azacitidine | NA | LDAC | Entinostat | WHO Classification |
| Burnett, 2013 [27] | 2004–2006 2007–2010 | G1: 76 (61–90) G2: 75 (54–86) | 495 | 40 | 195/495 (39.4) | LDAC | NA | LDAC | GO | Not specified |
| Sekeres, 2013 [29] | 2007–2010 | G1: 71 (60–87) G2: 70 (60–90) | 211 | 16 | 111/121 (52.6) | LDAC | NA | LDAC | Lintuzumab | WHO Classification |
| Kantarjian, 2012 [35] | 2006–2009 | G1: 73 (64–91) G2: 73 (65–86) | 457 | 24 | 197/485 (40.6) | LDAC | NA | Decitabine | NA | Bone marrow blast |
| Burnett, 2011 [26] | 2007–2009 | 74 (36–89) | 166 | 18 | 63/167 (37.9) | LDAC | NA | LDAC | ATO | Not specified |
| Fenaux, 2010 [25] | 2003–2007 | G1: 67 (61–89) G2: 70 (62–87) | 34 | 20.1 | 35/113 (30.9) | Azacitidine | NA | LDAC | NA | Not specified |
| **Non-Randomized studies** | | | | | | | | | | |
| Talati, 2020 [45] | 1995–2016 | 75.6 (70–97.5) | 346 | 20.5 | 117/346 (33.8) | HMA | NA | LDAC | NA | Not specified |
| Kanakasetty, 2019 [42] | 2013–2017 | G1: 68 (62–74) G2: 64 (61–74) | 139 | 15 | 69/188 (36.7) | LDAC | NA | HMA | NA | Immunophenotypically |
| Di Nardo 2019 [44] | 2014–2017 | 74 (56–86) | 145 | 15.4 | 64/145 (44) | Decitabine | Venetoclax | Azacitidine | Venetoclax | WHO Classification |
| Di Nardo, 2018 [41] | 2014–2016 | 75 (71–80) | 45 | 12.4 | 25/45 (55.5) | Decitabine | Venetoclax | Azacitidine | Venetoclax | WHO Classification |

*(Continued)*

**Table 1.** (Continued)

| Author, year | Time frame | Overall Age (y) Median, (Range) | Sample Size | Follow-up, Months (median) | Female Gender n/ N (%) | Main therapy | 2nd agent | Comparison therapy | 2nd agent | AML diagnosis |
|---|---|---|---|---|---|---|---|---|---|---|
| Boddu, 2017 [38] | 1990–2015 | 68 (60–75) | 406 | 6 | Not reported | HMA | NR | LDAC | NR | Bone marrow blast |
| Nanah, 2017 [43] | 2007–2015 | 76 (59–91) | 56 | 45 | 18/56 (32.1) | Azacitidine | NR | Decitabine | NR | WHO Classification |
| Jacob, 2015 [36] | 2011–2014 | G1: 75 (65–87) G2: 75 (60–91) | 30 | 8.7 | 6/30 (20) | Decitabine | NA | LDAC | NA | Bone marrow blast |
| Smith. 2014 [40] | 2006–2012 | G1:70.3 (11.8)[1] G2:69.4 (11.6)[1] | 487 | 30 | 279/487 (57.2) | Azacitidine | NA | Decitabine | NR | Not specified |
| Quintas-Cardama, 2012 [39] | 2000–2010 | G1 74 (65–84) G2: 73 (65–86) | 114 | 54 | 36/114 (31.6) | Azacitidine | HDI | Decitabine | HDI | Not specified |
| Di Febo, 2007 [51] | 1987–2003 | G1 67 (61–89) G2: 70 (62–87) | 28 | 3.3 | 15/28 (53.5) | LDAC | NA | LDAC | ATRA | Bone marrow blast |

LDAC, *low-dose* cytarabine, NA, not applicable, GO, Gemtuzumab ozogamicin, ATO, Arsenic trioxide, NR, not reported. HDI, Histone deacetykase inhibitors, HMA, Hypomethylating agents, ATRA, All-trans retinoic acid, NCCN; National comprehensive Cancer Network.

G1, Main therapy, G2, Comparison therapy.

1, mean (Standard deviation).

## Effects of the interventions

We summarize the effects of the interventions and their associated certainty of the evidence by creating one table per outcome. Table 2 summarize the effect of the interventions on the overall survival of the participants, Table 3 summarizes the effect of the interventions on the infectious severe adverse events (CTC adverse effects grade 3 or higher), and Table 4 summarizes the effect of the interventions on the non- infectious severe adverse events (CTC adverse effects grade 3 or higher). S1 Table summarizes the effect of the interventions on 1-year mortality, 30-days mortality, complete remission and length of hospital stay, and S2 Table summarizes the certainty of evidence from the sub-group analyses.

## Overall survival (OS)

**Overall survival over the longest follow-up time.** Seventeen studies [twelve RCTs (2,618 patients) and five NRS (1,523 patients)] reported overall survival, with a median follow-up ranged from 6 to 30 months (Table 1) [24,25,29–32,35,37,38,40–42,44–49]. We identified three main drugs (azacitidine (AZA), decitabine (DEC) and low-dose cytarabine (LDAC)) used as monotherapy or in combination with other agents, and a total of 9 comparisons (LDAC monotherapy vs DEC monotherapy [35], AZA monotherapy vs LDAC monotherapy [24,25], AZA monotherapy vs AZA combination [32,48], LDAC monotherapy vs LDAC combination [29–31,49], DEC monotherapy vs DEC combination [37,46,47], AZA monotherapy vs DEC monotherapy [40], LDAC combination vs hypomethylating agents (HMAs) [38], DEC combination vs AZA combination [32,48], and LDAC monotherapy vs HMAs [42,45]. From the nine comparisons, one had moderate certainty evidence, and showed little or no difference on survival between AZA monotherapy and LDAC monotherapy (HR 0.69, 95% CI 0.31–1.53, N = 2 RCTs, 346 patients, I2 56%) (Table 1, Fig 2) [24,25]. We identified four comparisons

**Table 2. Overall survival: Classification of the interventions based on paired meta-analysis for older adults with AML not candidate for intensive therapy.**

| Interventions (Follow-up; median range) | Relative effects and source | Absolute effects estimates | | Plain summary |
|---|---|---|---|---|
| | | Baseline risk for control group (per 1000) | Difference (95% CI) (per 1000) | |
| **High certainty (moderate to high certainty of evidence)** | | | | |
| AZAM vs LDACM[1] (20.1–24.4 months) [24,25] | HR 0.69, 95% CI 0.31–1.53, based on 346 patients in 2 RCTs. | 630 per 1000 | 134 fewer per 1000 (From 365 fewer to 152 more) | AZAM compared to LDACM probably has little or no effect on mortality. |
| **Low certainty (low to very low certainty of evidence)** | | | | |
| DECM vs DECC[2] (25–30 months) [37,46,47] | HR 0.85, 95% CI 0.65–1.10, based on 679 patients In 3 RCTs | 524 per 1000 | 56 fewer per 1000 (From 141 fewer to 34 more) | DECM compared to DECC may have little or no effect on mortality |
| LDACM vs LDACC[3] (16–28.2 months) [29–31,49] | HR 1.41, 95% CI 0.98–2.04, based on 620 patients in 4 RCTs. | 665 per 1000 | 121 more per 1000 (From 7 fewer to 228 more) | LDACM compared to LDACC may have little or no effect on mortality. |
| AZAM vs AZAC[4] (10–20.5 months) [32,48] | HR 1.33, 95% CI 0.96–1.85, based on 488 patients in 2 RCTs. | 717 per 1000 | 96 more per 1000 (From 15 fewer to 186 more) | AZAM compared to AZAC may have little or no effect on mortality. |
| LDACM vs DECM[5] (24 months) [35] | RR 0.99, 95% CI 0.91–1.08, based on 485 patients in 1 RCT | 814 per 1000 | 8 more per 1000 (From 73 fewer to 65 more) | LDACM compared to DECM may have little or no effect on mortality. |
| **AZAM vs DECM[6] (30 months) [40]** | **HR 0.72, 95% CI 0.57–0.92, based on 487 patients in 1 NRS.** | **623 per 1000** | **118 fewer per 1000 (from 196 fewer to 32 fewer)** | **AZAM compared to DECM may reduce mortality, however, we are very uncertain about this effect.** |
| **LDACM vs HMA[7] (15 to 20.5 months) [42,45]** | **HR 0.46, 95% CI 0.36–0.59, based on 485 patients in 2 NRS** | **518 per 1000** | **221 more per 100 (From 160 more to 271 more)** | **LDACM compared to HMA may increase mortality, however, we are very uncertain about this effect.** |
| **LDACC vs HMA[8] (6 months) [38]** | **RR 1.11, 95% CI 1.04–1.18, based on 406 patients in 1 NRS** | **850 per 1000** | **94 more per 1000 (from 34 more to 153 more)** | **LDACC compared to HMA may increase mortality, however, we are very uncertain about this effect.** |
| DECC vs AZAC[9] (15.1 months) [44] | Narrative description in the footnote, information based on 145 patients in 1 NRS. | NA[10] | NA[10] | DECC compared to AZAC may not have little or no effect on mortality however, we are very uncertain about this effect. |

Baseline risk information came from control group from the included studies. HR; hazard ratio, RR, relative risk, RCT, Randomized controlled studies, NRS, Non-randomized trials.

1. Moderate ⊕⊕⊕◯; Rate down by one level: serious imprecision (Effect estimate is not consistent with benefits and harms).

2. Low ⊕⊕◯◯: Rate down by two levels: serious inconsistency (I2 60%) and serious imprecision (effect estimate is not consistent with benefits and harms).

3. Low ⊕⊕◯◯: Rate down by two levels: serious inconsistency (I2 74%) and serious imprecision (effect estimate is not consistent with benefits and harms).

4. Low ⊕⊕◯◯: Rate down by two levels: serious inconsistency (I2 50%) and serious imprecision (effect estimate is not consistent with benefits and harms).

5. Low ⊕⊕◯◯: Rate down by two levels: Very serious imprecision (Effect estimate comes from a single study and is not consistent with benefits and harms).

6. Very low ⊕◯◯◯: Rate down by one level: Serious risk of bias (confounding factors in the results were not approached).

7. Very low ⊕◯◯◯: Rate down by one level: Serious risk of bias (confounding factors in the results were not approached).

8. Very low ⊕◯◯◯: Rate down by one level: Serious risk of bias (confounding factors in the results were not approached).

9. Very low ⊕◯◯◯: Rate down by one level: Serious risk of bias (confounding factors in the results were not approached).

10. DECC vs AZAC: Azacitidine plus Venetoclax show the following overall survival: 1200 mg; 8.8 95% CI 0.9—NR; 800 mg; 15.2 95% CI 9.1—NR; 400 mg; NR 95% CI NR 9.0—NR. Decitabine plus Venetoclax show the following overall survival: 1200 mg: NR 95% CI 12.4-NR; 800 mg; 17.5 95% CI 10.3-NR; 400 mg: 14.2 95% CI 7.7—NR.

with low certainty of evidence, that may have little or no effect on the survival of the participants (LDAC monotherapy vs DEC monotherapy [35], and AZAM monotherapy vs AZAM combination [32,48]) (Table 1). All other comparisons were very low certainty evidence,

**Table 3. GRADE summary of findings–infectious adverse events: Monotherapy or combination antileukemic therapy for older adults with AML not candidate for intensive therapy, evidence from randomized control studies and non-randomized studies.**

| Comparisons | Relative effects and source of evidence | Absolute effect estimates | | Certainty/Quality of evidence | Plain languages summary |
|---|---|---|---|---|---|
| | | Baseline risk for control group (per 1000) | Difference (95% CI) (per 1000) | | |
| **Septic shock** | | | | | |
| AZAM vs LDACM [24] | RR 0.65 95% CI 0.16–2.55, based on 389 patients in 1 RCT. | 26 per 1000 | 9 fewer per 1000 (From 22 fewer to 41 more) | Low ⊕⊕○○ (Very serious imprecision)[1] | AZAM compared to LDACM may have little or no effect on septic shock. |
| AZAM vs AZAC [33] | RR 1.29 95% CI 0.39–4.26, based on 32 patients from 1 RCT. | 222 per 1000 | 64 more per 1000 (from 136 fewer to 724 more) | Very low ⊕⊕○○ (Serious risk of bias and very serious imprecision)[2] | AZAM compared to AZAC may have little or no effect on septic shock, however, we are very uncertain about this effect. |
| **Febrile neutropenia** | | | | | |
| DECM vs DECC [37,46,47] | RR 0.85 95% CI 0.65–1.10, based on 671 patients from 3 RCTS. | 159 per 1000 | 22 fewer per 1000 (from 53 fewer to 14 more) | Moderate ⊕⊕⊕○ (Serious imprecision)[3] | DECM compared to DECC probably have little or no effect on febrile neutropenia. |
| **AZAM vs AZAC [48]** | **RR 0.45 95% CI 0.31–0.65, based on 427 patients from 1 RCTs.** | **417 per 1000** | **229 fewer per 1000 (From 288 fewer to 146 fewer)** | **Moderate ⊕⊕⊕○ (Serious imprecision)[4]** | **AZAM compared to AZAC probably decreases febrile neutropenia.** |
| AZAM vs LDACM [24] | RR 0.93 95% CI 0.68–1.28, based on 389 patients from 1 RCT. | 301 per 1000 | 21 fewer per 1000 (from 96 fewer to 84 more) | Low ⊕⊕○○ (Very serious imprecision)[1] | AZAM compared to LDACM may have little or no effect on febrile neutropenia. |
| LDACM vs DECM [35] | RR 0.77 95% CI 0.57–1.04, based on 446 patients from 1 RCT. | 319 per 1000 | 73 fewer per 1000 (From 137 fewer to 13 more) | Low ⊕⊕○○ (Very serious imprecision)[1] | LDACM compared to DECM may have little or no effect on febrile neutropenia. |
| LDACM vs LDACC [29–31,49] | RR 0.64 95% CI 0.40–1.00, based on 868 patients from 4 RCTs. | 308 per 1000 | 111 fewer per 1000 (From 185 fewer to 0 more) | Low ⊕⊕○○ (Serious inconsistency and serious imprecision)[5] | LDACM compared to LDACC may have little or no effect on febrile neutropenia. |
| DECC vs AZAC [44] | RR 1.31 95% CI 0.90–1–92, based on 145 patients from 1 NRS. | 375 per 1000 | 120 more per 1000 (From 37 fewer to 345 more) | Very low ⊕○○○ (Very serious risk of bias and serious imprecision)[6] | DECC compared to AZAC may have little or no effect on febrile neutropenia, however, we are very uncertain about this estimate. |
| **Pneumonia** | | | | | |
| DECM vs DECC [37,46,47] | RR 1.02 95% CI 0.73–1.42, based on 671 patients in 3 RCTs. | 201 per 1000 | 4 more per 1000 (From 54 fewer to 84 more) | Moderate ⊕⊕⊕○ (Serious imprecision)[3] | DECM compared to DECC probably have little or no effect on pneumonia. |
| LDACM vs DECM [35] | RR 0.88 95% CI 0.60–1.27, based on 446 patients in 1 RCTs. | 214 per 1000 | 26 fewer per 1000 (From 86 fewer to 58 more) | Low ⊕⊕○○ (Very serious imprecision)[1] | LDACM compared to DECM may have little or no effect on pneumonia |
| AZAM vs LDACM [24] | RR 1.01 95% CI 0.66–1.53, based on 389 patients in 1 RCT. | 190 per 1000 | 2 more per 1000 (From 64 fewer to 100 more) | Low ⊕⊕○○ (Very serious imprecision)[1] | AZAM compared to LDACM may have little or no effect on pneumonia. |
| LDACM vs LDACC [29,31,49] | RR 0.60 95% CI 0.33–1.11, based on 507 patients in 3 RCTs. | 147 per 1000 | 59 fewer per 1000 (From 99 fewer to 16 more) | Low ⊕⊕○○ (Serious risk of bias and serious imprecision)[6] | LDACM compared to LDACC may have little or no effect on pneumonia. |
| AZAM vs AZAC [48] | RR 1.26 95% CI 0.87–1.82, based on 427 patients from 1 RCTs. | 198 per 1000 | 51 more per 1000 (From 26 fewer to 162 fewer) | Low ⊕⊕○○ (Very serious imprecision)[1] | AZAM compared to AZAC may have little or no effect on pneumonia. |

*(Continued)*

**Table 3.** (Continued)

| Comparisons | Relative effects and source of evidence | Absolute effect estimates | | Certainty/Quality of evidence | Plain languages summary |
|---|---|---|---|---|---|
| | | Baseline risk for control group (per 1000) | Difference (95% CI) (per 1000) | | |
| LDACM vs HMA [42] | RR 0.60 95% CI 0.28–1.29, based on 139 patients in 1 NRS. | 207 per 1000 | 83 fewer per 1000 (From 149 fewer to 60 more) | Very low ⊕◯◯◯ (Very serious risk of bias and serious imprecision)[7] | LDACM compared to HMA may have little or no effect on pneumonia, however, we are very uncertain about this estimate. |
| **Sepsis** | | | | | |
| LDACM vs LDACC [29,49] | RR 0.98 95% CI 0.48–2.01, based on 420 patients in 2 RCTs. | 68 per 1000 | 1 fewer per 1000 (From 35 fewer to 69 more) | Moderate ⊕⊕⊕◯ (Serious imprecision)[3] | LDACM compared to LDACC probably have little or no effect on sepsis. |
| DECM vs DECC [37] | RR 1.44 95% CI 0.71–2.90, based on 163 patients in 1 RCT. | 136 per 1000 | 60 more per 1000 (from 39 fewer to 258 more) | Low ⊕⊕◯◯ (Very serious imprecision)[1] | DECM compared to DECC may have little or no effect on sepsis |
| AZAM vs AZAC [48] | RR 1.74 95% CI 0.72–3.03, based on 427 patients from 1 RCTs. | 57 per 1000 | 27 more per 1000 (From 16 fewer to 115 more) | Low ⊕⊕◯◯ (Very serious imprecision)[1] | AZAM compared to AZAC may have little or no effect on sepsis. |
| DECC vs AZAC [41] | RR 0.32 95% CI 0.01–7.45, based on 45 patients in 1 NRS. | 45 per 1000 | 31 fewer per 1000 (From 45 fewer to 293 more) | Very low ⊕◯◯◯ (Very serious risk of bias and serious imprecision)[6] | DEEC compared to AZAC may have little or no effect on sepsis, however, we are very uncertain about this effect. |

Baseline risk was obtained from the control group from the included studies.

1. We decided to rate down two levels due to imprecision: effect estimate is not consistent with benefit or harm and effect estimate comes from a single study.

2. We decided to rate down two levels due to risk of bias and imprecision: Allocation concealment was not described; adaptive randomization based on results, increase likelihood to be predicted and effect estimate comes from a single study and effect estimate is not consistent with benefit and harm.

3. We decided to rate down by one level due to imprecision: effect estimate is not consistent with benefit or harm.

4. We decided to rate down by one level due to imprecision: effect estimate comes from a single study.

5. We decided to rate down two levels due to inconsistency and imprecision: I2 62% (p-value 0.05) and effect estimate is not consistent with benefit or harms.

6. We decided to rate down two levels due to risk of bias and imprecision: Some of the covariates were not equal distribute among the participants (e. g. Hydroxyurea before study initiation) and The interventions related to the second agent might influence the treatment in the comparisons; Different proportions of patients in each group received granulocyte colony-stimulating factor or prophylactic non-azole antifungal agents. Venetoclax dose could be modified according to toxicity and effect estimate is not consistent with benefit or harms.

7. We decided to rate down two levels due risk of bias and imprecision: Performance status is different between the treatments under comparison (ECOG 3; 35.8% vs 0%), intervention status is well defined but some aspects of the assignments of intervention status were determined retrospectively and not clear if switches in treatment happen or co-interventions, also not clear if this was adjusted in the analysis and effect estimate comes from a single study.

which means that we are very uncertain about the true comparative effect of the interventions (Table 1). There was important inconsistency in two comparisons (DEC monotherapy vs DEC combination [37,47], and LDAC monotherapy vs LDAC combination) [29–31,49], for which we conducted subgroup analyses (Subgroup analysis section).

**All-cause of mortality at 1 year.** Seven RCTs (1,511 patients) addressing three comparisons reported all-cause mortality as the proportion of patient who died at 1 year (AZAM monotherapy vs LDAC monotherapy [24], LDAC monotherapy vs LDAC combination [26–30], and LDAC monotherapy vs HMAs [42,45]). Two of the comparisons reported a reduction on mortality (AZA monotherapy vs LDAC monotherapy; [RR 0.78, 95% CI 0.64–0.94, N = 1 RCT, 312 patients] [24], and, LDAC monotherapy vs HMAs [RR 0.46, 95% CI 0.36–0.59, N = 2 NRS, 485 patients, I2 0%] [42,45]). However, the certainty of the evidence was low, and very low, respectively, which means that we are not certain about the true effect of the interventions (S1 Table).

**Table 4. GRADE summary of findings—Non-infectious severe adverse events: Monotherapy or combination antileukemic therapy for older adults with AML not candidate for intensive therapy, evidence from randomized control studies and non-randomized studies.**

| Comparisons | Relative effects and source of evidence | Absolute effect estimates | | Certainty/Quality of evidence | Plain languages summary |
|---|---|---|---|---|---|
| | | Baseline risk for control group (per 1000) | Difference (95% CI) (per 1000) | | |
| **Anemia** | | | | | |
| DECM vs DECC [46,47] | RR 0.85 95% CI 0.68–1.06, based on 512 patients from 2 RCTs. | 373 per 1000 | 56 fewer per 1000 (From 119 fewer to 22 more) | Moderate ⊕⊕⊕○ (Serious imprecision)[1] | DECM compared to DECC probably have little or no effect on anemia. |
| AZAM vs LDACM[24,25] | RR 0.79 95% CI 0.58–1.09, based on 421 patients from 2 RCTs. | 287 per 1000 | 60 fewer per 1000 (From 120 fewer to 26 more) | Moderate ⊕⊕⊕○ (Serious imprecision)[1] | AZAM compared to LDACM probably have little or no effect on anemia. |
| LDACM vs DECM [35] | RR 0.80 95% CI 0.60–1.07, based on 446 patients from 1 RCT. | 336 per 1000 | 67 fewer per 1000 (From 134 fewer to 24 more) | Low ⊕⊕○○ (Very serious imprecision)[2] | LDACM compared to DECM may have little or no effect on anemia. |
| LDACM vs LDACC [29,31,49] | RR 0.88 95% CI 0.55–1.39, based on 545 patients from 3 RCTs. | 240 per 1000 | 29 fewer per 1000 (From 108 fewer to 93 more) | Low ⊕⊕○○ (Serious inconsistency and serious imprecision)[3] | LDACM compared to LDACC may have little or no effect on anemia |
| AZAM vs AZAC [34,48] | RR 1.07 95% CI 0.78–1.46, based on 576 patients from 2 RCT. | 310 per 1000 | 27 more per 1000 (from 68 fewer to 143 more) | Low ⊕⊕○○ (Serious inconsistency and serious imprecision)[4] | AZAM compared to AZAC may have little or no effect on anemia. |
| DECC vs AZAC [44] | RR 1.29 95% CI 0.77–1.88, based on 145 patients from 1 NRS | 318 per 1000 | 92 more per 1000 (from 73 fewer to 280 more) | Very low ⊕○○○ (Very serious risk of bias and serious imprecision)[5] | DECC compared to AZAC may have little or no effect on anemia, however, we are very uncertain about this estimate. |
| **Neutropenia** | | | | | |
| **LDACM vs DECM [35]** | **RR 0.62 95% CI 0.44–0.86, based on 446 patients in 1 RCTs.** | **319 per 1000** | **121 fewer per 1000 (From 179 to 45 fewer)** | **Moderate ⊕⊕⊕○ (Serious imprecision)[6]** | **LDACM compared to DECM probably decreases neutropenia.** |
| DECM vs DECC [46,47] | RR 0.82 95% CI 0.64–1.06, based on 512 patients from 2 RCTs. | 310 per 1000 | 56 Fewer per 1000 (From 112 fewer to 19 more)[1] | Moderate ⊕⊕⊕○ (Serious imprecision)[1] | DECM compared to DECC probably have little or no effect on neutropenia. |
| AZAM vs LDACM [24,25] | RR 1.00 95% CI 0.81–1.24, based on 421 patients from 2 RCTs | 316 per 1000 | 0 per 1000 (From 60 fewer to 76 more) | Moderate ⊕⊕⊕○ (Serious imprecision)[1] | AZAM compared to LDACM probably have little or no effect on neutropenia. |
| LDACM vs LDACC [29,31,49] | RR 0.92 95% CI 0.28–3.08, based on 545 patients in 3 RCTs. | 278 per 1000 | 50 fewer per 1000 (From 195 fewer to 345 more) | Low ⊕⊕○○ (Serious inconsistency and serious imprecision)[3] | LDACM compared to LDACC may have little or no effect on neutropenia. |
| AZAM vs AZAC [34,48] | RR 0.84 95% CI 0.54–1.31, based on 576 patients from 2 RCTs. | 483 per 1000 | 77 fewer per 1000 (From 222 fewer to 150 more) | Low ⊕⊕○○ (Serious inconsistency and serious imprecision)[7] | AZAM compared to AZAC may have little or no effect on neutropenia. |
| DECC vs AZAC [41] | RR 1.29 95% CI 0.77–1.88, based on 45 patients from 1 NRS. | 318 per 1000 | 92 more per 1000 (From 73 fewer to 280 more) | Very low ⊕○○○ (Very serious risk of bias and serious imprecision)[4] | DECC compared to AZAC may have little or no effect on neutropenia, however, we are very uncertain about this estimate. |
| LDACM vs HMA [42] | RR 0.97 95% CI 0.77–1.22, based on 139 patients in 1 NRS. | 690 per 1000 | 21 fewer per 1000 (From 159 fewer to 152 more) | Very low ⊕○○○ (Very serious risk of bias and serious imprecision)[8] | LDACM compared to HMA may have little or no effect on neutropenia, however, we are very uncertain about this estimate. |
| **Thrombocytopenia** | | | | | |
| DECM vs DECC [46,47] | RR 0.92 95% CI 0.67–1.26, based on 512 patients from 2 RCTs. | 427 per 1000 | 34 Fewer per 1000 (From 141 fewer to 111 more)[1] | Moderate ⊕⊕⊕○ (Serious imprecision)[1] | DECM compared to DECC probably have little or no effect on thrombocytopenia. |

*(Continued)*

**Table 4.** (Continued)

| Comparisons | Relative effects and source of evidence | Absolute effect estimates | | Certainty/Quality of evidence | Plain languages summary |
|---|---|---|---|---|---|
| | | Baseline risk for control group (per 1000) | Difference (95% CI) (per 1000) | | |
| AZAM vs LDACM [24,25] | RR 0.92 95% CI 0.78–1.07, based on 422 patients from 2 RCTs | 351 per 1000 | 28 fewer per 1000 (From 77 fewer to 25 more) | Moderate ⊕⊕⊕◯ (Serious imprecision)[1] | AZAM compared to LDACM probably have little or no effect on thrombocytopenia. |
| AZAM vs AZAC [34,48] | RR 0.91 95 CI% 0.78–1.06, based on 576 patients from 2 RCTs. | 511 per 1000 | 46 fewer per 1000 (From 112 fewer to 31 more) | Moderate ⊕⊕⊕◯ (Serious imprecision)[1] | AZAM compared to AZAC probably have little or no effect on thrombocytopenia. |
| LDACM vs LDACC [29,31,49] | RR 0.86 95% CI 0.67–1.10, based on 545 patients in 3 RCTs. | 368 per 1000 | 52 fewer per 1000 (From 122 fewer to 37 more) | Moderate ⊕⊕⊕◯ (Serious imprecision)[1] | LDACM compared to LDACC probably have little or no effect on thrombocytopenia. |
| LDACM vs DECM [35] | RR 0.88 95% CI 0.69–1.12, based on 446 patients in 1 RCTs. | 399 per 1000 | 48 fewer per 1000 (From 124 to 48 more) | Low ⊕⊕◯◯ (Very serious imprecision)[2] | LDACM compared to DECM may have little or no effect on thrombocytopenia |
| LDACM vs HMA [42] | RR 1.17 95% CI 0.94–1.46, based on 139 patients in 1 NRS. | 655 per 1000 | 111 more per 1000 (From 39 fewer to 301 more) | Very low ⊕◯◯◯ (Very serious risk of bias and very serious imprecision)[8] | LDACM compared to HMA may have little or no effect on thrombocytopenia, however, we are very uncertain about this estimate. |
| DECC vs AZAC [41] | RR 0.43 95% CI 0.18–1.05, based on 45 patients from 1 NRS. | 500 per 1000 | 285 fewer per 1000 (From 410 fewer to 25 more) | Very low ⊕◯◯◯ (Very serious risk of bias and very serious imprecision)[5] | DECC compared to AZAC may have little or no effect on thrombocytopenia, however, we are very uncertain about this estimate. |
| **Hospitalization** | | | | | |
| **AZAM vs DECM [40]** | **RR 0.87 CI95% 0.76–0.99, based on 487 patients from 1 NRS.** | **709 per 1000** | **92 fewer per 1000 (From 170 fewer to 7 fewer)** | **Very low ⊕◯◯◯ (Serious risk of bias and very serious imprecision)[9]** | **AZAM compared to DECM may have little effect on hospitalization, however, we are very uncertain about this effect.** |
| **Hypoxia/Respiratory Failure** | | | | | |
| LDACM vs LDACC [30] | RR 0.19 95% CI 0.01–3.78, based on 87 patients in 1 RCTs. | 48 per 1000 | 39 fewer per 1000 (From 47 fewer to 132 more) | Low ⊕⊕◯◯ (Very serious imprecision)[10] | LDACM compared to LDACC may have little or no effect on Hypoxia/Respiratory Failure. |

Baseline risk was obtained from the control group from the included studies.

1. We decided to rate down one level due to imprecision: effect estimate is not consistent with benefit or harm.

2. We decided to rate down two levels due to imprecision: Effect estimate comes from single study and is not consistent with benefit or harm.

3. We decided to rate down two levels due to inconsistency and imprecision. I2 41% (p-value 0.18) and effect estimate is not consistent with benefit or harm.

4. We decided to rate down two levels due to serious inconsistency and imprecision; effect estimate not consistent with benefit or harm and I2 of 45%.

5. We decided to rate down two levels due to risk of bias and imprecision. Some of the covariates were not equal distribute among the participants (e. g. Hydroxyurea before study initiation) and The interventions related to the second agent might influence the treatment in the comparisons; Different proportions of patients in each group received granulocyte colony-stimulating factor or prophylactic non-azole antifungal agents. Venetoclax dose could be modified according to toxicity and effect estimate is not consistent with benefit or harms.

6. We decided to rate down one level due to imprecision; effect estimate come from a single study.

7. We decided to rate down two levels due to inconsistency and imprecision: I2 84% (p-value 0.01) and effect estimate is not consistent with benefit and harm.

8. We decided to rate down by two levels due to risk of bias and imprecision. Confounding expected due to imbalance in the compared groups. (Performance status is different between the treatments under comparison (ECOG: 3; 35.8% versus 0%) and the adherence to the intended intervention is not clear and effect estimate come from a single study which is not consistent with benefit or harms.

9. We decided to rate down by one level due to risk of bias. Researchers did not account for relevant prognostic factors in the results and effect estimate comes from a single study.

10. We decided to rate down two levels due to imprecision; effect estimate come from a single study, effect estimate is not consistent with benefit and harms and small event rate.

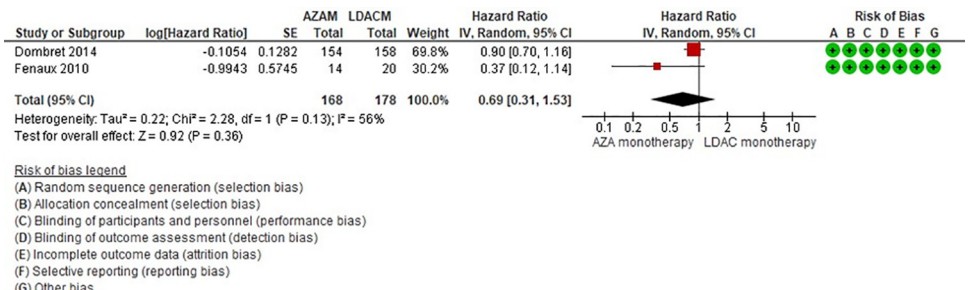

**Fig 2. Overall survival between azacitidine monotherapy vs low-dose cytarabine monotherapy.**

**All-cause of mortality at 30 days.** Seven RCTs (1,334 patients), addressing two comparisons reported all-cause mortality as the proportion of patient who died at 30 days (DEC monotherapy vs DEC plus bortezomib [37], and, LDAC monotherapy vs LDAC combination [26–28,30,31,49]). The comparisons suggested have little or no difference on patient mortality at 30 days. However, the certainty of the evidence was low (S1 Table).

## Infectious adverse events (AEs)

**Septic shock.** Two RCTs (421 patients) addressing two comparisons reported septic shock (AZAM monotherapy vs LDAC monotherapy [24], and, AZA monotherapy vs AZA plus vorinostat [33]). The comparisons suggested little or no difference in the development of septic shock. However, the certainty of the evidence was low, and very low respectively (Table 3).

**Febrile neutropenia.** Ten RCTs (2,801 patients) and one NRS (145 patients) addressing 6 comparisons reported febrile neutropenia events (LDAC monotherapy vs DEC monotherapy [33], AZA monotherapy vs LDAC monotherapy [24], LDAC monotherapy vs LDAC combination [29–31,49], DEC monotherapy vs DEC combination [37,46,47], AZA monotherapy vs AZA plus venetoclax [48], and DEC plus venetoclax vs AZA plus venetoclax [44]). Two of the six were moderate certainty evidence. When comparing AZA monotherapy vs AZA plus venetoclax, patients treated with AZA monotherapy had a lower risk of fewer febrile neutropenia events (RR 0.45, 95% CI 0.31–0.65, N = 1 RCT, 427 participants) [48], and when comparing DEC monotherapy vs DEC combination (RR 0.85, 95% CI 0.65–1.10, N = 3 RCTs, 671 patients, I2 0%) [37,46,47] (Fig 3) there was probably little or no difference in the risk of febrile neutropenia. Four comparisons have low (AZA monotherapy vs LDAC monotherapy [24],

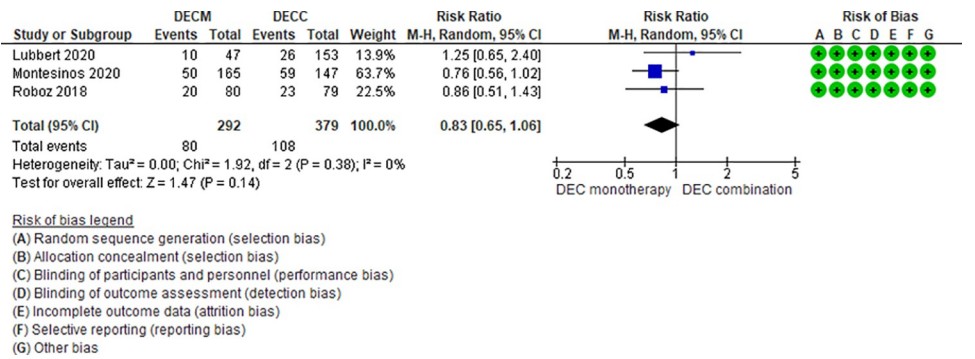

**Fig 3. Febrile neutropenia events between decitabine monotherapy vs decitabine combination.**

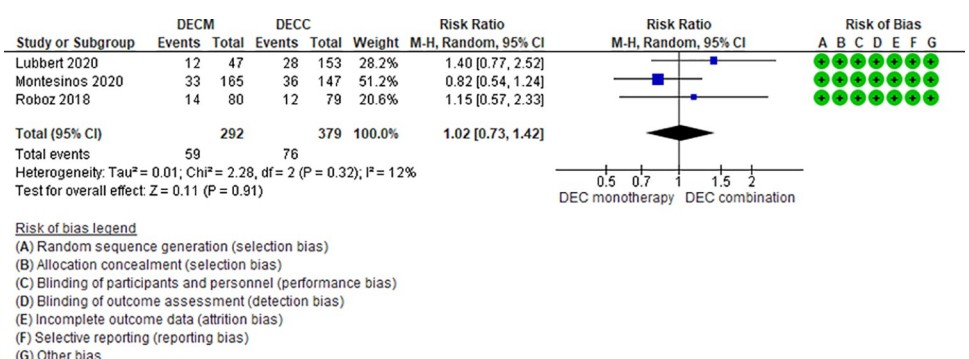

**Fig 4. Pneumonia events between decitabine monotherapy vs decitabine combination.**

LDAC monotherapy vs DEC monotherapy [35], and LDAC monotherapy vs LDAC combination [29–31,49]) and very low (DEC plus venetoclax vs AZA plus venetoclax [44]) certainty of evidence (Table 3).

**Pneumonia.** Nine RCTs (2,440 patients) and one NRS (139 patients) addressing 6 comparisons reported the presence of pneumonia (LDAC monotherapy vs DEC monotherapy [35], AZA monotherapy vs LDAC monotherapy [24], DEC monotherapy vs DEC combination [37,46,47], LDAC monotherapy vs LDAC combination [29,31,49], AZAM monotherapy vs AZA plus venetoclax [48], and LDAC monotherapy vs HMAs [42]). One of the six comparisons is moderate certainty evidence. When comparing DEC monotherapy vs DEC combination there is probably little or no difference in the risk of pneumonia (RR 1.02, 95% CI 0.73–1.42, N = 3 RCTs, 671 patients, I2 12%) (Fig 4) [37,46,47]. Five comparisons are low (LDAC monotherapy vs DEC monotherapy [35], AZA monotherapy vs LDAC monotherapy [24], LDAC monotherapy vs LDAC combination [29,31,49], and AZA monotherapy vs AZA plus venetoclax [48]) and very low (LDAC monotherapy vs HMAs [42]) certainty of evidence (Table 3).

**Sepsis.** Four RCTs (1,010 patients) and one NRS (45 patients) addressing four comparisons reported sepsis (LDAC monotherapy vs LDAC combination [29,49], DEC monotherapy vs DEC plus bortezomib [37], AZAM monotherapy vs AZA plus venetoclax [48], and DEC plus venetoclax vs AZA plus venetoclax [41]). One of the four is moderate certainty evidence. When comparing LDAC monotherapy vs LDAC combination there is probably little or no difference in the risk of sepsis (RR 0.98, 95% CI 0.48–2.01, N = 2 RCTs, 420 patients, I2 0%) (Fig 5) [29,49]. Three comparisons are low (DEC monotherapy vs DEC plus bortezomib [37],

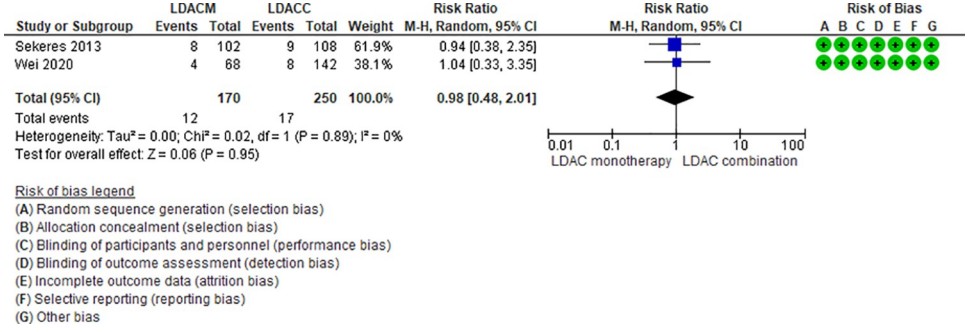

**Fig 5. Sepsis events between low-dose cytarabine monotherapy vs low-dose cytarabine combination.**

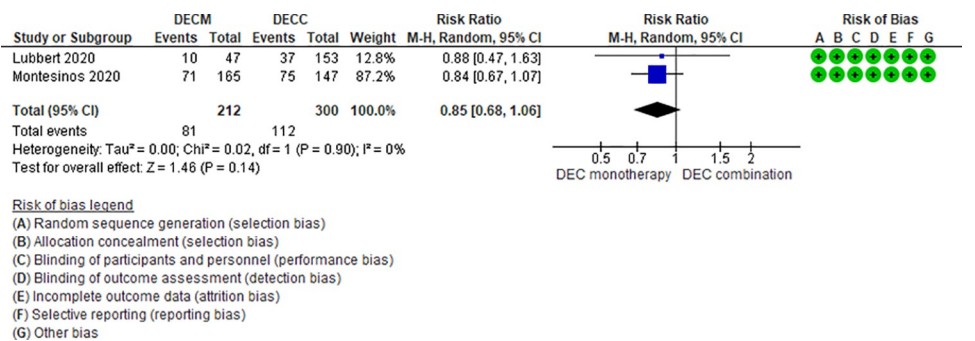

**Fig 6. Anemia events between decitabine monotherapy vs decitabine combination.**

and AZA monotherapy vs AZA plus venetoclax [48]) and very low (DEC plus venetoclax vs AZA plus venetoclax [41]) certainty of evidence (Table 3).

### Non-infectious adverse events (AEs)

**Anemia.** Ten RCTs (2,500 patients) and one NRS (145 patients) addressing 6 comparisons reported anemia (DEC monotherapy vs DEC combination [46,47], AZA monotherapy vs LDAC monotherapy [24,25], LDAC monotherapy vs DEC monotherapy [35], LDAC monotherapy vs LDAC combination [29,31,49], AZA monotherapy vs AZA combination [32,48], and, DEC plus venetoclax vs AZA plus venetoclax [44]). Two of the six were moderate certainty evidence. When comparing DEC monotherapy vs DEC combination (RR 0.85, 95% CI 0.68–1.06, N = 2 RCTs, 512 patients, I2 0%) (Fig 6) [46,47], and AZA monotherapy vs LDAC monotherapy there is probably little or no difference in the risk of anemia (RR 0.79, 95% CI 0.58–1.09, N = 2 RCTs, 512 patients, I2 17%) (Fig 7) [24,25]. Four comparisons are low (LDAC monotherapy vs DEC monotherapy [35], LDAC monotherapy vs LDAC combination [29,31,49], and AZA monotherapy vs AZA combination [34,48]) and very low (DEC combination vs AZA combination [44]) certainty of evidence (Table 4).

**Neutropenia.** Ten RCTs (2,500 patients) and two NRS (184 patients) addressing seven comparisons reported neutropenia (DEC monotherapy vs DEC combination [46,47], AZA monotherapy vs LDAC monotherapy [24,25], LDAC monotherapy vs DEC monotherapy [35], LDAC monotherapy vs LDAC combination [29,31,49], AZA monotherapy vs AZA combination [34,48], DEC combination vs AZA combination [41], and LDAC monotherapy vs HMAs [42]). Three of those seven are moderate certainty evidence. When comparing LDAC monotherapy vs DEC monotherapy, patients treated with LDAC monotherapy shown fewer

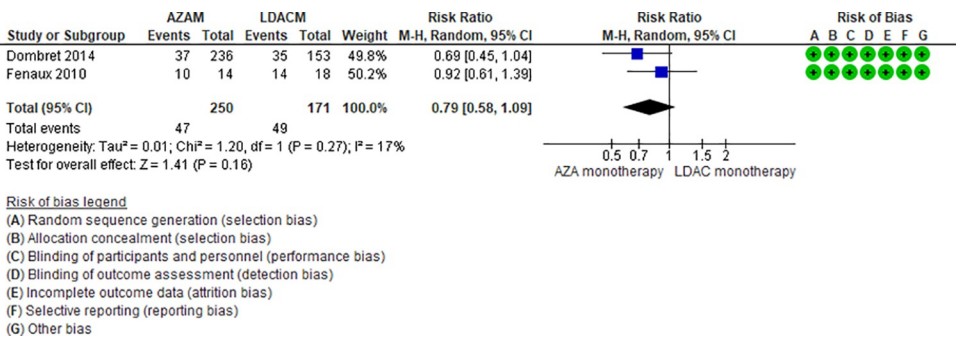

**Fig 7. Anemia events between azacitidine monotherapy vs low-dose cytarabine monotherapy.**

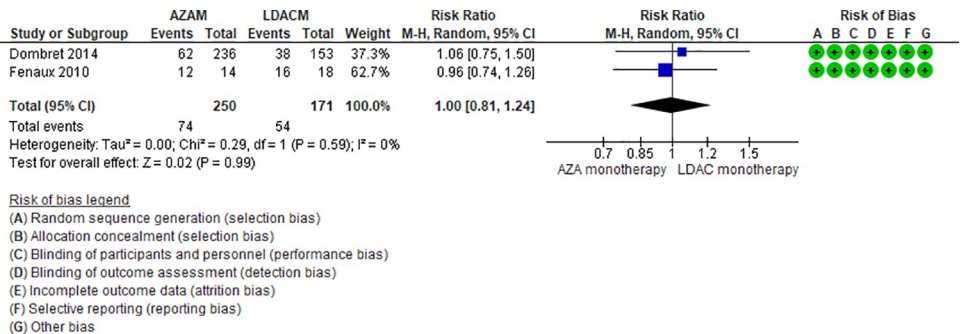

**Fig 8. Neutropenia events between azacitidine monotherapy vs low-dose cytarabine monotherapy.**

neutropenia events (RR 0.62, 95% CI 0.44–0.86, N = 1 RTC, 446 patients) [35], and when comparing AZA monotherapy vs LDAC monotherapy (RR 1.00, 95% CI 0.81–1.24, N = 2 RCTs, 421 patients, I2 0%) (Fig 8) [24,25] and DEC monotherapy vs DEC combination (RR 0.82, 95% CI 0.64–1.06, N = 2 RCTs, 512 patients, I2 0%) (Fig 9) ([46,47] there is probably little or no difference in the risk of neutropenia. Four comparisons are low (LDAC monotherapy vs LDAC combination [29,31,49], and AZA monotherapy vs AZA combination [34,48]) and very low (DEC combination vs AZA combination [41], and LDAC monotherapy vs HMAs [42]) certainty of evidence (Table 4).

**Thrombocytopenia.** Ten RCTs (2,500 patients) and two NRS (184 patients) addressing seven comparisons reported thrombocytopenia (DEC monotherapy vs DEC combination [46,47], AZA monotherapy vs LDAC monotherapy [24,25], LDAC monotherapy vs DEC monotherapy (35), LDAC monotherapy vs LDAC combination [29,31,49], AZA monotherapy vs AZA combination [34,48], DEC combination vs AZA combination [41], and LDAC monotherapy vs HMAs [42]). Four of these seven are moderate certainty evidence. When comparing DEC monotherapy vs DEC combination (RR 0.92, 95% CI 0.67–1.23, N = 2 RCTs, 512 patients, I2 34%) (Fig 10) [46,47], AZA monotherapy vs LDAC monotherapy (RR 0.92, 95% CI 0.78–1.07, N = 2 RCTs, 422 patients, I2 0%) (Fig 11) [24,25], AZA monotherapy vs AZA combination (RR 0.91, 95% CI 0.78–1.06, N = 2 RTC, 576 patients, I2 0%) (Fig 12) [34,48], and LDAC monotherapy vs LDAC combination (RR 0.86, 95% CI 0.67–1.10, N = 3 RCTs, 545 patients, I2 0%) (Fig 13) [29,31,49], there is probably little or no difference in the risk of thrombocytopenia. Four comparisons are low (LDAC monotherapy vs DEC monotherapy [35]) and

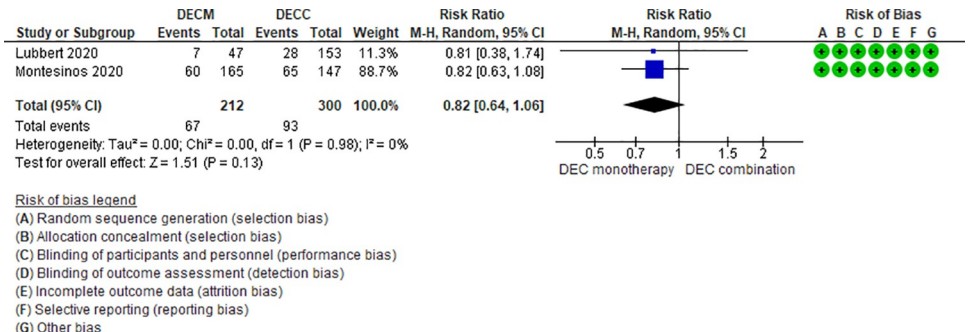

**Fig 9. Neutropenia events between decitabine monotherapy vs decitabine combination.**

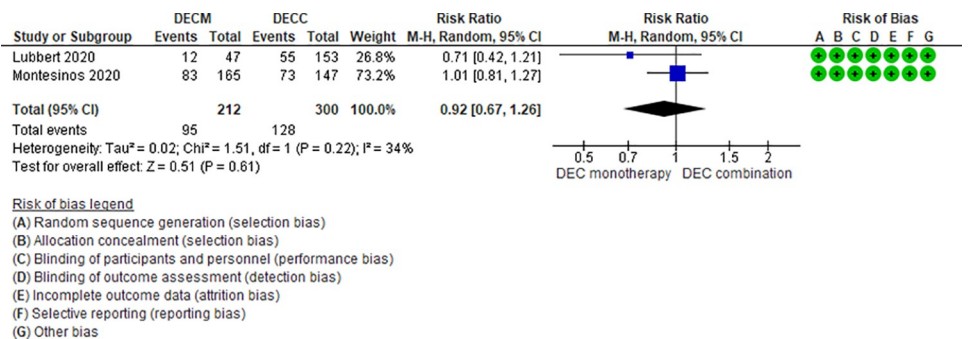

**Fig 10. Thrombocytopenia between decitabine monotherapy vs decitabine combination.**

very low (DEC combination vs AZA combination [41], and LDAC monotherapy vs HMAs [42]) certainty of evidence (Table 4).

**Hospitalization and hypoxia.** One NRS (478 patients) [40] and one RCT (87 patients) [30] addressing two comparisons reported on hospitalization (very low certainty evidence) and hypoxia/respiratory failure (low certainty evidence). When comparing LDAC monotherapy vs LDAC combination no difference was found in hypoxia/respiratory failure development. When comparing AZA monotherapy vs DEC monotherapy, fewer hospitalizations occurred in favor of AZA monotherapy. However, we are very uncertain about this effect (Table 4).

## Other outcomes

**Complete remission over the longest follow-up.** 5 RCTs (1,331 patients) and 1 NRS (114 patients)] addressing three comparisons reported complete remission as event-free survival (AZA monotherapy vs AZA combination [48], LDAC monotherapy vs LDAC combination [26–28,49], and, AZA monotherapy vs DEC monotherapy [39]). One of these is moderate certainty of evidence. When comparing AZA monotherapy vs AZA combination, patients treated with AZA monotherapy shown a decrease in the event-free survival (HR 1.59, 95% CI 1.26–2.00, N = 1 RCTs, 488 patients) [48]. One comparison is very low certainty (AZA monotherapy vs DEC monotherapy [37]) (S1 Table). There was important inconsistency in one comparison (LDAC monotherapy vs LDAC combination) [26–28,49], which we conducted subgroup analyses (Subgroup analysis section).

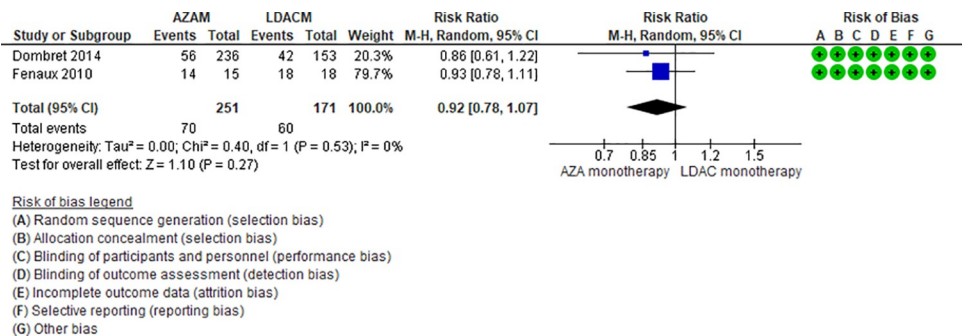

**Fig 11. Thrombocytopenia between azacitidine monotherapy vs low-dose cytarabine monotherapy.**

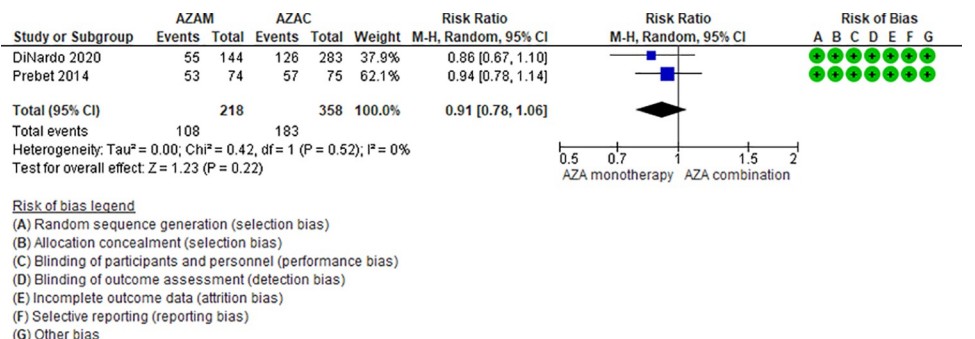

**Fig 12. Thrombocytopenia between azacitidine monotherapy vs azacitidine combination.**

**Length of hospital stay.** Two RCTs (598 patients) addressing one comparison (LDAC monotherapy vs LDAC combination) reported the length of hospital stay [27,28]. When comparing both drugs (MD 8.24 days, 95% CI -18.71 to 2.24, N = 2 RCTs, 598 patients, I2 83%) there is little or no effect on the length of hospital stay (S1 Table).

## Subgroup and sensitivity analysis

The included studies did not provide sufficient information to performed a sensitivity analysis base on the risk of bias. We observed important inconsistency in two comparisons from two outcomes: Overall survival (DEC monotherapy vs DEC combination [37,46,47], and LDAC monotherapy vs LDAC combination [29–31,49]), and 12-month relapse-free survival (LDAC monotherapy vs LDAC combination [26–28,49]), for which we conducted subgroup analyses based on the secondary agent of the combination

**DEC monotherapy vs DEC combination.** We identified five secondary agents from three RCTs (679 patients) reporting overall survival [37,46,47]. All the comparisons are low certainty of evidence. Talacotuzumab (HR 1.04, 95% 0.79–1.37, N = 1 RCT, 316 patients) [47], Bortezomib (HR 1.17, 95% CI 0.84–1.63, N = 1 RCT, 163 patients) [37], and Valproate (HR 0.85, 95% CI 0.57–1.27, N = 1 RCT arm) [46] has little or no effect in the overall survival of participants compared to DEC monotherapy. When comparing all-trans retinoic acid (HR 0.58, 95% CI 0.37–0.91, N = 1 RCT arm) and all-trans retinoic acid plus valproate (HR 0.62, 95% CI 0.40–0.96, N = 1 RCT arm) against DEC monotherapy, patients treated with the combination

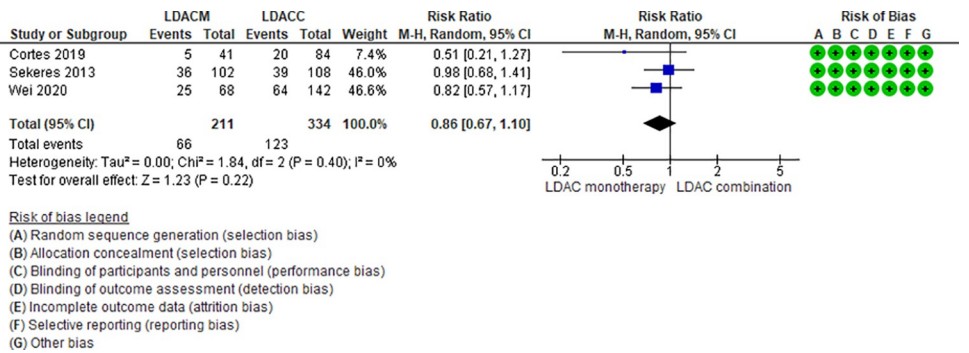

**Fig 13. Thrombocytopenia between low-dose cytarabine monotherapy vs low-dose cytarabine combination.**

therapy shown higher overall survival (S1 Fig) [46]. However, we are uncertain about the true effect of these comparisons (S2 Table).

**LDAC monotherapy vs LDAC combination.** Overall survival. We identified four secondary agents from four RCTs (620 participants) reporting overall survival [29–31,49]. All the comparisons are low certainty of evidence. Venetoclax (HR 1.33, 95% CI 0.92–1.92, N = 1 RCT, 211 patients) [49], and Lintuzumab (HR 0.95, 95% CI 0.72–1.25, N = 1 RCT, 211 patients) has little or no effect in the overall survival of participants compared to LDAC monotherapy [29]. When comparing volasertib (HR 1.59, 95% CI 1.00–2.52, N = 1 RCT, 87 patients) [30] and glasdegib (HR 2.17, 95% CI 1.44–3.26, N = 1 RCT, 111 patients) [31], patients treated with combination therapy shown higher overall survival (S2 Fig). However, we are uncertain about the true effect of these comparisons (S2 Table).

Complete remission. We identified four secondary agents from four RCTs (843 patients) reporting the 12-month relapse-free survival. All the comparisons are low certainty of evidence. Gemtuzumab ozogamicin plus LDAC against LDAC monotherapy (HR 1.11, 95% CI 0.73–1.69, N = 1 RCT, 494 participants) has little or no effect in the 12-month relapse-free survival [27]. When comparing LDAC plus arsenic trioxide against LDAC monotherapy (HR 2.95, 95% CI 1.21–7.19, N = 1 RCT, 34 participants), we found an improve of the 12-month relapse-free survival on patients treated with LDAC monotherapy [26], and when comparing vosaroxin plus LDAC (HR 0.41, 95% CI 0.16–1.02, N = 1 RCT, 104 participants) [28] and venetoclax plus LDAC (HR 0.58, 95% CI 0.42–0.80, N = 1 RCT, 211 patients) [49] we found an improvement of the 12-month relapse-free survival on patients treated with LDAC combination therapy (S3 Fig). However, we are uncertain about the true effect of these comparisons (S3 Table).

## Discussion

The elderly population diagnosed with AML who are not candidates for intensive antileukemic therapy propose an important challenge. In the last two decades' new therapeutic options have become available with a reasonable effectiveness and excellent toxicity profile. However, uncertainty remains about the comparative effectiveness and safety of the different available options. In order to help clinicians and patients during the decision-making process, we summarize the best available evidence by conducting a systematic review with several meta-analyses.

### Summary of the evidence

Our systematic review identified three main drugs (azacitidine, decitabine and low-dose cytarabine), as monotherapies or in combination, addressing nine comparisons. We found information on patients´ OS, 1-year mortality, 30-days' mortality, infectious and non-infectious AEs, complete remission and length of hospital stay. We found no evidence regarding quality of life, functional status and burden of caregiver for any comparison.

Most of the evidence comes from RCTs (3,902 patients). However, due to the small number of patients per comparison (imprecision), and inconsistency between the treatment effects reported by different studies, most of the evidence was judged as low or very low certainty. Evidence about the effects on OS was available for all nine comparisons, with no compelling evidence in favor of any of the available options. There is moderate certainty in one of the comparisons (AZA monotherapy vs LDAC monotherapy), showing little no differences in the OS between the patients treated with these drugs. We performed two subgroup analyses for this outcome (DEC monotherapy vs DEC combination, and LDAC monotherapy vs LDAC combination). Also, we performed another subgroup analysis for the complete remission

outcome (LDAC monotherapy and LDAC combination). Overall, we found single studies with favorable effects in combination therapy groups (LDAC combination and, DEC combination). However, due to the number of studies, the sample size, and the inconsistency between the pooled estimates, we classified the evidence as low certainty (Table 2). The evidence for other outcomes and comparisons was scarce and we could not conduct more of these analyses.

Toxicity is a very important feature during the decision-making process. We observed a similar prevalence of severe adverse events (CTC grade 3 or higher), except for two. AZA combination therapy (venetoclax) had more febrile neutropenia events when compared against AZA monotherapy (Table 3), and DEC monotherapy had more neutropenia events when compared against LDAC monotherapy (Table 4).

## Strengths and limitations

No prior SRs addressed alternative chemotherapy for older patients with AML in whom intensive therapy was not an option. We conducted a comprehensive database search; specified explicit eligibility criteria; and conducted duplicate, independent study selection, data extraction and risk of bias assessment with resolution of disagreement with discussion and third-party adjudication where necessary. We used the GRADE approach to assess the quality of the evidence for NRS and RCT studies and where informative included both relative and absolute effects. We included all the relevant options that either RCTs or NRS had addressed.

We faced an important challenge when conducting our meta-analysis: The secondary agents varied across the studies within each comparison and, for most of the comparisons the type of secondary agent was not the same. We decided to pool studies within the comparisons regardless the secondary agent, and to explore if the secondary agent was associated with the treatment effect when comparing monotherapies vs. combination therapies. During the clinical practice guideline development, we planned additional analyses based on the input from the panel members. Unfortunately, the number of studies within comparisons and outcomes was insufficient to conduct such analyses. With the available evidence when developing the recommendations, the panel believed that any extra analyses, including sensitivity analyses that would exclude specific studies (e.g., diagnostic criteria for AML), was unlikely to change their conclusions. Also, we planned to performed a network meta-analysis (NMA) to compare all interventions against each other. At the end of data extraction, we identified insufficient evidence to do so (data not shown). This decision created the challenge to summarize all the useful evidence across the nine comparisons; we provided a summary on main text but also provide extensive supplementary information in the appendices.

## Implications

Treating older AML patients can be challenging, as clinicians and patients must balance the goal of increasing longevity with the risk that more aggressive treatment may increase adverse events and hospitalization. During the recommendation formulation process, with the evidence available at that time, the guideline panel found no compelling evidence of additional benefit with more aggressive treatment with more than one agent, and instances in which such therapy did increase adverse events. After the meeting, however, some new studies (RCTs and NRS) reported benefits of combinations over monotherapy, for example, DEC combined with ATRA and VPA+ATRA may result in better survival than DEC monotherapy [Lubbert 2020] [46], and AZA combined with venetoclax may also result in better survival than AZA monotherapy [DiNardo 2020] [48]. Because these results were inconsistent with the previously identified studies, when including these new studies in the meta-analyses, the certainty of the

overall evidence decreased. It is important to notice, however, that the certainty of evidence for each of these specific comparisons is low.

Therapy selection for older adults with AML who are not candidates for intensive antileukemic therapy is based on the patient fitness, patients' characteristics (cytogenic and molecular profiles), the trade-off between drug safety and toxicity, and patients' values and preferences [52]. The scientific community agrees on offering therapies based on HMA agents (e.g., azacytidine, decitabine) with some exceptions: liver and kidney severe disease, prior HMA therapy, and the presence of an actionable mutation [52,53]. For these populations other options are available (e.g., Low-dose cytarabine). Currently, combination therapy has become the standard of care for unfit AML older patients. However, the secondary agent depends on their availability in each setting and the presence of specific genetic mutations. Venetoclax (BCL2 inhibitor) is the preferred secondary agent to add to the HMA therapies, this is based on promising results from NRS and RCTs (mentioned previously). In our review, we identify benefits from the combination therapy with venetoclax. However, the certainty of the effect was judged to be low after creating a pooled estimate (imprecision and inconsistency). The same situation was identified with other secondary agents. We are aware that creating pooled estimates without stratifying based on the second agent may impact the effect estimate of a specific agent (e.g., venetoclax). In the comparison with enough studies, we undertook a subgroup analysis to explore their effect. However, the AZA monotherapy vs AZA combination did not have sufficient studies to explore it.

Our evidence suggests HMA therapies are acceptable options with similar efficacy and safety to other less-intensive treatment options. The certainty of the evidence was, however, low for most comparisons and outcomes, and there was no published evidence for several outcomes considered critical for decision-making. The limitations of the evidence also highlight the need for additional randomized trials including a wider range of patient-important outcomes–in particular quality of life—to definitively establish the relative merits of alternative regimens in older patients with AML in whom more aggressive therapy is not an option.

## Supporting information

**S1 Checklist. PRISMA 2020 checklists.**
(DOCX)

**S1 Fig. Overall survival subgroup analysis.** Decitabine monotherapy vs decitabine combination.
(TIF)

**S2 Fig. Overall survival subgroup analysis.** Low-dose cytarabine monotherapy vs low dose cytarabine combination.
(TIF)

**S3 Fig. Complete remission subgroup analysis.** Low-dose cytarabine monotherapy vs low dose cytarabine combination.
(TIF)

**S1 Table. Summary of findings table for 1-year mortality, 30-day mortality, complete remission and length of hospital stay.**
(DOCX)

**S2 Table. Overall survival subgroup analyses–summary of findings table.**
(DOCX)

**S3 Table. Relapse free survival subgroup analyses–summary of findings table.**
(DOCX)

**S1 Appendix. Eligibility criteria and study characteristics.**
(DOCX)

**S2 Appendix. Search strategy items.**
(DOCX)

**S3 Appendix. Study characteristics.**
(DOCX)

**S4 Appendix. Risk of bias of the included studies.**
(DOCX)

## Acknowledgments

We thank the clinical experts, Desai Pinkal, Mark Litzow, Mikkael A. Sekeres for critical feedback on population and outcome selection, and treatment grouping.

## Author Contributions

**Conceptualization:** Pinkal Desai, Mark Litzow, Mikkael A. Sekeres, Gordon H. Guyatt, Romina Brignardello-Petersen.

**Data curation:** Luis Enrique Colunga-Lozano, Fernando Kenji Nampo, Arnav Agarwal, Romina Brignardello-Petersen.

**Formal analysis:** Luis Enrique Colunga-Lozano.

**Methodology:** Gordon H. Guyatt, Romina Brignardello-Petersen.

**Project administration:** Luis Enrique Colunga-Lozano.

**Supervision:** Romina Brignardello-Petersen.

**Visualization:** Luis Enrique Colunga-Lozano.

**Writing – original draft:** Luis Enrique Colunga-Lozano.

**Writing – review & editing:** Luis Enrique Colunga-Lozano, Fernando Kenji Nampo, Arnav Agarwal, Pinkal Desai, Mark Litzow, Mikkael A. Sekeres, Gordon H. Guyatt, Romina Brignardello-Petersen.

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
