## [Decision Letter · Decision Letter 0]

1 Jul 2021

PONE-D-21-13321

Less intensive anti-leukemic therapies (monotherapy and/or combination) for older adults with acute myeloid leukemia who are not candidates for intensive anti-leukemic therapy: a systematic review and meta-analysis.

PLOS ONE

Dear Dr. Colunga-Lozano,

Thank you for submitting your manuscript to PLOS ONE. After careful consideration, we feel that it has merit but does not fully meet PLOS ONE’s publication criteria as it currently stands. Therefore, we invite you to submit a revised version of the manuscript that addresses the points raised during the review process.

We look forward to receiving your revised manuscript.

Kind regards,

Mona Pathak, PhD

Academic Editor

PLOS ONE

Additional Editor Comments:

Please revise as per reviewer's suggestions.

Journal Requirements:

2. Please described the data extraction methods in more details. We would expect to see reporting of the specific information extracted from the manuscripts.

3. We noted in your submission details that a portion of your manuscript may have been presented or published elsewhere.

[Yes

This systematic review contributed to development of the 2020 ASH guidelines for treating newly diagnosed older adults with AML.

We consider that is worth it to publish the review of this topic, with more details than the information provided in the guideline.]

Please clarify whether this publication was peer-reviewed and formally published. If this work was previously peer-reviewed and published, in the cover letter please provide the reason that this work does not constitute dual publication and should be included in the current manuscript.

[The authors have declared that no competing interests exist related to this work.

During the development of the clinical practice guidance, the American Society of Hematology deal with the competing interests. Information added as "Related Manuscript"].

6. Please include captions for ALL your Supporting Information files at the end of your manuscript, and update any in-text citations to match accordingly. Please see our Supporting Information guidelines for more information: http://journals.plos.org/plosone/s/supporting-information.

Reviewers' comments:

Reviewer's Responses to Questions

**Comments to the Author**

1. Is the manuscript technically sound, and do the data support the conclusions?

Reviewer #1: Yes

Reviewer #2: Partly

2. Has the statistical analysis been performed appropriately and rigorously? 

Reviewer #1: No

Reviewer #2: Yes

3. Have the authors made all data underlying the findings in their manuscript fully available?

Reviewer #1: No

Reviewer #2: Yes

4. Is the manuscript presented in an intelligible fashion and written in standard English?

Reviewer #1: Yes

Reviewer #2: No

5. Review Comments to the Author

Reviewer #1: Authors have attempted a very critical hypothesis to reveal the effect of Less intensive anti-leukemic therapies (monotherapy and/or combination) for older adults with acute myeloid leukemia who are not candidates for intensive anti-leukemic therapy by conducting a systematic review and meta-analysis. However, in the process of the manuscript review, following points were noticed and raised for clarification and valid answers.

Comments

1. Is the protocol for the current review published or registered at the protocol registries such as PROSPERO or Cochrane collaboration or other registries? Kindly shed light on this.

2. Information on the places where these studies took place, follow-up duration is good to make available in Table 1 or supplementary material.

3. Studies citations in all the tables are a must wherever the included studies were reported.

4. The number of studies should be made available under specific electronic search databases such as Medline and Embase at the first stage in Figure 1S. The number of duplicate studies utilizing both the databases is also important to be available in the same figure 1S. The authors have claimed in Figure 1S that the additional records through other sources are zero. But under the result section, the authors have mentioned that the two of the studies were provided by the clinical experts. Such a number must come in the PRISMA flowchart i.e. Figure 1S.

5. Authors have mentioned in the final stage of the PRISMA flowchart that the 27 studies were included for qualitative synthesis. Such a statement needs to be modified since the authors have also conducted the quantitative synthesis in addition to qualitative synthesis.

6. Is the total number of 27 studies included in those two studies which were provided by the clinical expert? Kindly shed light on this.

7. Why did the authors not plan for sensitivity analysis? Sensitivity analysis must be available by excluding few studies such as Burnett- 2011; Burnett- 2013; and Fenaux- 2010 studies under RCT as the AML diagnosis is not specified in a randomized controlled trial. Similarly, Talati- 2020; and Quintas-Cardam a – 2012 may be excluded for NRS studies for sensitivity analysis. The other criteria could be other diagnostic criteria for AML; single center vs multicentric studies; conventional trial vs clustered-randomized trial; the region where studies conducted (North-America; Asia, Europe, etc.), separate analysis for randomized trial and observational studies, shorter vs. longer follow-up, etc. Detail reporting of sensitivity analysis will ensure the stability of the computed effect sizes.

8. It would be nice to supplement the results with the number of studies on each of the 9 comparisons (number of studies for LDACM vs DECM, number of studies for AZAM vs LDACM, etc.) on any difference in overall survival in Table 2. Similarly the presentation of the results with number of studies in other outcome measures (various adverse effects) would be appreciated.

9. Authors made a plan to conduct subgroup analysis under the method section but the same is not available in the result section. What were the considered criteria for subgroup analysis? Kindly make available the subgroup analysis results in the light of the risk of bias assessment in the effect estimates.

10. The authors declared that the heterogeneity was assessed through Q statistic and I-square under the statistical section but the same is not explained in the results. Results may be needed to discuss in the light of heterogeneity. Subgroup analysis needs to be performed on the overall survival when at least a moderate level of heterogeneity was observed.

Addressing the above comments in the manuscript would be highly appreciated to enhance the overall manuscript quality and for the final outcome.

Kind regards

Bhaskar

Reviewer #2: Dear Authors,

You aim to present the results of a systematic review of utility (efficacy) of monotherapy vs. combination therapy for elderly newly diagnosed AML subjects who are unfit for intensive chemotherapeutic regimen.

Following are my observations:

1. The topic is of genuine importance for day to day clinical practice.

2.ABSTRACT:

#Could have been made more clear and concise.

#Objective is not clearly written.....To compare what with what?

# Results: Data can be presented in a simplified manner for better understanding (95% CI)

3. Short running title: needs to be short...can be "Less intensive chemotherapy in elderly AML"

4. Manuscript seems bit incoherent with complex sentence formatting making the things difficult to comprehend.....with

improper reference citation both in the text and in the final list.

5. When so many abbreviations were used it is better to present a glossary of all those at one place.

6. Main observations:

a)In two of the trials venetoclax has been added with azacytidine and decitabine . Combination Venetoclax therapy has

been shown to consistently improve OS significantly as compared to single agent hypomethylating agents. However in

results it was only mentioned about cytarabine , azacytidine and decitabine without mentioning the targeted agents.

b)The statistics part is well written .no mention of relapse and death rate is mentioned

c) Targeted agents were not taken into consideration

SINCERELY,

REVIEWERs

6. PLOS authors have the option to publish the peer review history of their article (what does this mean?). If published, this will include your full peer review and any attached files.

Reviewer #1: **Yes: **Bhaskar Thakur

Reviewer #2: **Yes: **Dr Somanath Padhi, M.D., All India Institute of Medical Sciences, Bhubaneswar, India

---

## [Author Response · Author response to Decision Letter 0]

31 Aug 2021

30th August 2021

Dr. Mona Pathak, PhD

Academic Editor

Plos One

Dear Dr. Pathak,

Thank you for taking the time to review our manuscript ID PONE-D-21-13321, and for all the thoughtful feedback. Please see below our responses (in red) to the comments regarding our manuscript, “Less intensive antileukemic therapies (monotherapy and/or combination) for older adults with acute myeloid leukemia who are not candidates for intensive antileukemic therapy: a systematic review and meta-analysis”. All changes to the manuscript are tracked.

Please let us know if anything requires further clarification.

Sincerely,

Luis Enrique Colunga Lozano

On behalf of the authors

Additional editor’s comments

We adjusted the format in the manuscript and files names according to PLOS ONE´s style requirements. We made the following changes:

- We modified the manuscript file name.

- We modified the names of the figures (Fig 1 and so forth) 

- We added the figures´ captions after the first paragraph in which the figure was cited. 

- We modified the names of the supporting information files

- We added, verified, and double-check the text and tables citations. 

- We added new Level 3 headings in the results sections. 

 2. Please describe the data extraction methods in more details. We would expect to see reporting of the specific information extracted from the manuscripts.

We adjusted the manuscript to describe with more detail the information requested. We added the following paragraph. 

Pairs of reviewers independently abstracted data on a standardized form. We extracted the following information: type of study, recruitment time-frame, follow-up (months), sample size, participant characteristics, as age (years), gender, cytogenetics (intermediate or poor), performance status (ECOG or WHO classification), white cell count, AML diagnosis criteria, trial location, source of funding, trial registry interventions (main agent, dose and second agent for combination therapy groups), comparisons (main agent, dose and second agent for combination therapy groups), and outcomes (mortality, quality of life, functional status, recurrence, morphologic complete remission, severe toxicity (CTC adverse effects grade 3 or higher), or burden on caregivers, at any time point. If reviewers could not resolve disagreement through discussion, a third reviewer adjudicated (RBP). 

3. We noted in your submission details that a portion of your manuscript may have been presented or published elsewhere. Please clarify whether this publication was peer-reviewed and formally published. If this work was previously peer-reviewed and published, in the cover letter please provide the reason that this work does not constitute dual publication and should be included in the current manuscript.

Our response is also included in the covert letter. 

This systematic review was performed to inform the American Society of Hematology (ASH) guideline for the treatment of older adults with acute myeloid leukemia. The manuscript was peer reviewed and published in the journal Blood Advances, on august 2020 (Mikkael A. Sekeres, et al; American Society of Hematology 2020 guidelines for treating newly diagnosed acute myeloid leukemia in older adults. Blood Adv 2020; 4 (15): 3528–3549), and included as supplementary material some tables that summarized the evidence. We consider that our manuscript does not constitute a dual publication. The amount of information overlapping between the guideline and this manuscript is not substantial, because the guideline does not provide details about the systematic review and meta-analysis process performed for each of the recommendations. In addition, for this manuscript we updated the search date and included new information not considered during the guideline development process. As requested, we have described this in the cover letter.

Our response is also included in the covert letter. 

We can confirm that “This does not alter our adherence to PLOS ONE policies on sharing data and materials.”

Our response is also included in the covert letter. 

We can confirm that the authors involved in this research work do not have any competing interest to declare. 

5. We note that you have indicated that data from this study are available upon request. PLOS only allows data to be available upon request if there are legal or ethical restrictions on sharing data publicly.

Our response is also included in the covert letter. 

We changed this section to “no restriction in sharing data”, as there are no legal or ethical restrictions. We included data to replicate our findings in the supporting information files. 

6. Please include captions for ALL your Supporting Information files at the end of your manuscript, and update any in-text citations to match accordingly

We included captions for all the supporting information files and we verified and update the in-text citations. 

Reviewer comments.

Reviewer #1: Authors have attempted a very critical hypothesis to reveal the effect of Less intensive anti-leukemic therapies (monotherapy and/or combination) for older adults with acute myeloid leukemia who are not candidates for intensive anti-leukemic therapy by conducting a systematic review and meta-analysis. However, in the process of the manuscript review, following points were noticed and raised for clarification and valid answers.

Comments

1. Is the protocol for the current review published or registered at the protocol registries such as PROSPERO or Cochrane collaboration or other registries? Kindly shed light on this

Unfortunately, no. The systematic review was not registered in PROSPERO or other registries, as the project was done within the American Society of Hematology methodology. We modified the current section by providing a statement regarding this situation, which reads:

“This systematic review was not registered on PROSPERO or other registries. This systematic review was performed with ASH guideline methodology (9) and informed the development of recommendations regarding the treatment of AML in elderly patients from the ASH 2020 Guidelines for treating newly diagnosed acute myeloid Leukemia in Older adults (8). The eligibility criteria for studies to include were pre-established by the panel when formulating the recommendation questions. We conducted the study in accordance with the Cochrane handbook (10) and report the results according to the Preferred Addressing Items for Systematic Reviews and Meta-Analyses guidelines (11) (S1 checklist)”. 

2. Information on the places where these studies took place, follow-up duration is good to make available in Table 1 or supplementary material.

Thanks a lot for the suggestion. The follow-up was available in table 1, row number 5 – Follow-up, months (median). We made the following changes:

- We removed the trials´ location available in the text. 

- We include a detailed description of the trials´ location in the S4 Appendix. 

3. Studies citations in all the tables are a must wherever the included studies were reported.

Thanks for bringing this situation to our attention. We made the following changes:

- References were added to the tables. 

- All the references in the manuscript were double-checked. 

4. The number of studies should be made available under specific electronic search databases such as Medline and Embase at the first stage in Figure 1S. The number of duplicate studies utilizing both the databases is also important to be available in the same figure 1S. The authors have claimed in Figure 1S that the additional records through other sources are zero. But under the result section, the authors have mentioned that the two of the studies were provided by the clinical experts. Such a number must come in the PRISMA flowchart i.e. Figure 1S

Thanks for bringing this situation to our attention. We updated our search strategy and included your suggestion before submitting our revised manuscript. We made the following changes:

- We modified the Figure 1: Records identified from databases (n=15,615) and other sources (n=2). Records remove before screening (n= 3,239), records screened (n=12,376), records excluded (n= 12,230), records assessed for eligibility (n=149), full-text studies excluded (n= 122) and studies included in the review (n=27).

- We added the requested numbers in the S3 appendix. We identify 8,576 records in Ovid Medline and 7,039 records in Embase.

5. Authors have mentioned in the final stage of the PRISMA flowchart that the 27 studies were included for qualitative synthesis. Such a statement needs to be modified since the authors have also conducted the quantitative synthesis in addition to qualitative synthesis.

We agree with the suggestion. We made the following changes: 

- We modified Fig 1 to report the includes studies as “Studies included in the review”.

6. Is the total number of 27 studies included in those two studies which were provided by the clinical expert? Kindly shed light on this.

Thanks for bringing this situation to our attention. We update our search strategy before submitting our revised version. We made the following changes:

After the removal of duplicates, we identified 12,376 studies of which 149 proved to be potentially relevant based on title an abstract screening. After full text review, we included 27 studies (Fig 1). From the included studies, 21 were included after the first search and informed the development of the recommendations (24-43), 6 studies were included after the guideline recommendation (44-49). We did not find any ongoing studies. 

7. Why did the authors not plan for sensitivity analysis? Sensitivity analysis must be available by excluding few studies such as Burnett- 2011; Burnett- 2013; and Fenaux- 2010 studies under RCT as the AML diagnosis is not specified in a randomized controlled trial. Similarly, Talati- 2020; and Quintas-Cardam a – 2012 may be excluded for NRS studies for sensitivity analysis. The other criteria could be other diagnostic criteria for AML; single center vs multicentric studies; conventional trial vs clustered-randomized trial; the region where studies conducted (North-America; Asia, Europe, etc.), separate analysis for randomized trial and observational studies, shorter vs. longer follow-up, etc. Detail reporting of sensitivity analysis will ensure the stability of the computed effect sizes.

Thanks for bringing this situation to our attention. We planned additional analyses based on the input of the guideline panel, who believed that the exclusion of some specific studies or characteristics was unlikely to change their conclusions and thus did not request a sensitivity or subgroup analysis. For most of the comparisons, we identified a small number of studies, which did not provide enough information to perform a robust analysis. In our current analysis, we included two comparisons with enough information to perform a sub-group analyses based on the second agent (LDAC monotherapy vs LDAC combination, and DEC monotherapy vs DEC combination). We made the following changes. 

- We modified the sub-group and sensitivity analysis section

We pooled and reported results from RCTs and NRS separately. We planned to conduct sensitivity analyses to explore the impact of the risk of bias in the effect estimates. We performed a subgroup analysis to explore the impact of the secondary agent (when comparing a combination therapy group) in the effect estimates, when there were sufficient studies.

- We added the sub-group analysis in the results 

Subgroup and sensitivity analysis

The included studies did not provide sufficient information to performed a sensitivity analysis base on the risk of bias. We observed important inconsistency in two comparisons from two outcomes: Overall survival (DEC monotherapy vs DEC combination (37, 46, 47), and LDAC monotherapy vs LDAC combination (29-31,49) (29-31, 49)), and 12-month relapse-free survival (LDAC monotherapy vs LDAC combination (26-28, 49)), for which we conducted subgroup analyses based on the secondary agent of the combination

DEC monotherapy vs DEC combination. 

We identified five secondary agents from three RCTs (679 patients) reporting overall survival (37, 46, 47). All the comparisons are low certainty of evidence. Talacotuzumab (HR 1.04, 95% 0.79 – 1.37, N= 1 RCT, 316 patients) (47), Bortezomib (HR 1.17, 95% CI 0.84 – 1.63, N= 1 RCT, 163 patients) (37), and Valproate (HR 0.85, 95% CI 0.57 – 1.27, N= 1 RCT arm) (46) has little or no effect in the overall survival of participants compared to DEC monotherapy. When comparing all-trans retinoic acid (HR 0.58, 95% CI 0.37 – 0.91, N= 1 RCT arm) and all-trans retinoic acid plus valproate (HR 0.62, 95% CI 0.40 – 0.96, N=1 RCT arm) against DEC monotherapy, patients treated with the combination therapy shown higher overall survival (S7 Figure) (46). However, we are uncertain about the true effect of these comparisons (S8 Table).

LDAC monotherapy vs LDAC combination. 

Overall survival. We identified four secondary agents from four RCTs (620 participants) reporting overall survival (29-31, 49). All the comparisons are low certainty of evidence. Venetoclax (HR 1.33, 95% CI 0.92 – 1.92, N= 1 RCT, 211 patients) (49), and Lintuzumab (HR 0.95, 95% CI 0.72 – 1.25, N= 1 RCT, 211 patients) has little or no effect in the overall survival of participants compared to LDAC monotherapy (29). When comparing volasertib (HR 1.59, 95% CI 1.00 – 2.52, N = 1 RCT, 87 patients) (30) and glasdegib (HR 2.17, 95% CI 1.44 – 3.26, N= 1 RCT, 111 patients) (31), patients treated with combination therapy shown higher overall survival (S9 Figure). However, we are uncertain about the true effect of these comparisons (S8 Table).

Complete remission. We identified four secondary agents from four RCTs (843 patients) reporting the 12-month relapse-free survival. All the comparisons are low certainty of evidence. Gemtuzumab ozogamicin plus LDAC against LDAC monotherapy (HR 1.11, 95% CI 0.73 – 1.69, N= 1 RCT, 494 participants) has little or no effect in the 12-month relapse-free survival (27). When comparing LDAC plus arsenic trioxide against LDAC monotherapy (HR 2.95, 95% CI 1.21 – 7.19, N=1 RCT, 34 participants), we found an improve of the 12-month relapse-free survival on patients treated with LDAC monotherapy (26), and when comparing vosaroxin plus LDAC (HR 0.41, 95% CI 0.16 – 1.02, N= 1 RCT, 104 participants) (28) and venetoclax plus LDAC (HR 0.58, 95% CI 0.42 - 0.80, N = 1 RCT, 211 patients) (49) we found an improvement of the 12-month relapse-free survival on patients treated with LDAC combination therapy (S10 Figure). However, we are uncertain about the true effect of these comparisons (S11 Table).

- We added two paragraphs in the discussion section.

We faced an important challenge when conducting meta-analysis, as the secondary agents varied across the studies within each comparison and for most of the comparisons the type of secondary agent was not the same. We decided to pool studies within comparisons regardless the secondary agent, and to explore if the secondary agent was associated with the treatment effect when comparing monotherapies vs. combinations.

8. It would be nice to supplement the results with the number of studies on each of the 9 comparisons (number of studies for LDACM vs DECM, number of studies for AZAM vs LDACM, etc.) on any difference in overall survival in Table 2. Similarly, the presentation of the results with number of studies in other outcome measures (various adverse effects) would be appreciated.

We agree with the suggestion. We made the following changes: 

- We added in the text a detail description for the survival and mortality outcomes at different follow-ups.

- We added two summaries of findings tables to describe the adverse events.

- We modified the text within each outcome to provide more details about the number of studies and patients per comparison.

9. Authors made a plan to conduct subgroup analysis under the method section but the same is not available in the result section. What were the considered criteria for subgroup analysis? Kindly make available the subgroup analysis results in the light of the risk of bias assessment in the effect estimates.

Thanks for bringing this situation to our attention. Subgroup analysis section was added to the results section. 

10. The authors declared that the heterogeneity was assessed through Q statistic and I-square under the statistical section but the same is not explained in the results. Results may be needed to discuss in the light of heterogeneity. Subgroup analysis needs to be performed on the overall survival when at least a moderate level of heterogeneity was observed.

Thanks for bringing this situation to our attention. We provided a detailed description of the inconsistency between the comparisons in the footnotes from each "summary of findings table". As reported in the previous comment, we added a sub-group analysis for those comparisons with moderate to high inconsistency and enough information to perform a robust analysis. We made the following changes:

- We moved two "summary of findings tables" from the supporting information to the main text. 

- We added the I2% in the text description for comparisons with moderate certainty of evidence.

- We added subgroup analysis in the result section (previously mentioned)

- We added the following paragraph in the discussion section.

During the recommendation formulation process, with the evidence available at that time, the guideline panel found no compelling evidence of additional benefit with more aggressive treatment with more than one agent, and instances in which such therapy did increase adverse events. After the meeting, however, some new studies (RCTs and NRS) reported benefits of combinations over monotherapy, for example, DEC combined with ATRA and VPA+ATRA combination may result in better survival than DEC monotherapy [Lubbert 2020] AZA combined with venetoclax may also result in better than AZA monotherapy [DiNardo 2020]. Because these results were inconsistent with the previously identified studies, when including these new studies in the meta-analyses, the certainty of the overall evidence decreased. It is important to notice, however, that the certainty of evidence for each of these specific comparisons is low.

Reviewer #2: Dear Authors,

You aim to present the results of a systematic review of utility (efficacy) of monotherapy vs. combination therapy for elderly newly diagnosed AML subjects who are unfit for intensive chemotherapeutic regimen.

Following are my observations:

1. The topic is of genuine importance for day to day clinical practice.

2.ABSTRACT:

#Could have been made more clear and concise.

. We agree with the suggestion. We modified the abstract structure to be more clear and concise. We made the following modifications.

- We reduce the introduction.

- We modified the objective, which reads as “To compare the effectiveness and safety of less intensive antileukemic therapies for older adults with newly diagnosed AML not candidates for intensive therapies”.

- We added a statistical analysis description, which reads as We calculated pooled hazard ratios (HRs), risk ratios (RRs), mean differences (MD) and their 95% confidence intervals (CIs) using a random-effects pairwise meta-analyses and assessed the certainty of evidence using the Grading of Recommendations Assessment, Development, and Evaluation (GRADE) approach”

- We modified the results section to show the relative effects instead of the absolute effects. 

- We arrange the sentence structure. 

- We remove the funding statement.

#Objective is not clearly written.....To compare what with what?

Thanks for bringing this situation to our attention. The objective was modified to be more clear. Now is describe as “We conducted a systematic review to address compared the comparative effectiveness and safety of low-intensity antileukemic therapies (monotherapy and/or combination) in older adults with newly diagnosed AML who are not candidates for intensive therapy.

# Results: Data can be presented in a simplified manner for better understanding (95% CI)

We agree with the suggestion. We made the following modifications to the results section.

- We change the absolute effects to relative effects in all the comparisons. 

3. Short running title: needs to be short...can be "Less intensive chemotherapy in elderly AML"

We agree with the suggestion. We modified the short title to “Less intensive chemotherapy in elderly AML

4. Manuscript seems bit incoherent with complex sentence formatting making the things difficult to comprehend.....with improper reference citation both in the text and in the final list.

Thanks for bringing this situation to our attention. We made the following modifications.

- We change most of the sentence to active voice.

- We modified our abbreviations to be less complex, by only abbreviating the main comparison drug. 

- We double checked all the references.

5. When so many abbreviations were used it is better to present a glossary of all those at one place.

Thanks for bringing this situation to our attention. The Plos one submission guidance does not provide a section to report abbreviations. We modified the result section to remove the complex abbreviations. As describe previously we abbreviate the main comparison drugs: Azacitidine (AZA), Decitabine (DEC), and Low-dose cytarabine (LDAC). 

6. Main observations:

a) In two of the trials venetoclax has been added with azacitidine and decitabine. Combination Venetoclax therapy has been shown to consistently improve OS significantly as compared to single agent hypomethylating agents. However, in results it was only mentioned about cytarabine, azacitidine and decitabine without mentioning the targeted agents.

Thanks for bringing this situation to our attention. As you mentioned, we identify different target agents, for example, venetoclax was used in two non-randomized studies (Di Nardo 2018 and Di Nardo 2019) and two RCTs (Wei 2020 and Di Nardo 2020) with different comparisons (Di Nardo 2018 and 2019 – Decitabine + venetoclax vs Azacitidine + venetoclax, Di Nardo 2020 – Azacitidine + venetoclax vs Azacitidine monotherapy, and Wei 2020 – LDAC + venetoclax vs LDAC monotherapy). We agree on the importance of the target agents and their respective descriptions. Due to the amount of information available, we only describe in detail the secondary agents from the comparisons with statistical inconsistency. We made the following changes.

- In the result section, we added the second agent when we report the effect estimates from a single trial. 

- we added a detail description of the target agents used in each study within each comparison. Which read as follows.

We identified 9 comparisons from the 27 included studies: two parallel RCTs (433 patients) compared azacitidine monotherapy against low-dose cytarabine monotherapy (24, 25); four parallel RTCs (921 patients) compared azacitidine monotherapy against azacitidine in combination with a second agent (venetoclax (48), entinostat (34) and vorinostat (32, 33)); three NRS (648 patients) compared azacitidine monotherapy against decitabine monotherapy (39, 40, 50); three parallel RCTs (685 participants) compared decitabine monotherapy against decitabine in combination with a second agent (bortezomib(37), valproate and/or retinoic acid (46) and talacotuzumab(47)); two NRS (190 patients) compared decitabine in combination with a second agent (venetoclax) against azacitidine in combination with a second agent (venetoclax) (41, 44); seven parallel RCTs (1406 patients) and one NRS (28 patients) compared low-dose cytarabine monotherapy against low-dose cytarabine in combination with a second agent (ATRA (51), arsenic trioxide (26), gemtuzumab ozogamicin (27), lintuzumab (29), volasertib (30), vosaroxin (28), glasdegib (31), and venetoclax (49)); one parallel RCT (457 patients) and one NRS (30 patients) compared low-dose cytarabine monotherapy against decitabine monotherapy (35, 36); one NRS (406 patients) compared low-dose cytarabine in combination with a second agent (not specified) against hypomethylating agents(38); and, two NRS (485 patients) compared low-dose cytarabine monotherapy against hypomethylating agents (42, 45).

We included three sub-group analysis describing the effect of the target agents within the comparisons with enough information to performed a robust analysis. 

Subgroup and sensitivity analysis

The included studies did not provide sufficient information to performed a sensitivity analysis base on the risk of bias. We observed important inconsistency in two comparisons from two outcomes: Overall survival (DEC monotherapy vs DEC combination (37, 46, 47), and LDAC monotherapy vs LDAC combination (29-31,49) (29-31, 49)), and 12-month relapse-free survival (LDAC monotherapy vs LDAC combination (26-28, 49)), for which we conducted subgroup analyses based on the secondary agent of the combination

DEC monotherapy vs DEC combination. 

We identified five secondary agents from three RCTs (679 patients) reporting overall survival (37, 46, 47). All the comparisons are low certainty of evidence. Talacotuzumab (HR 1.04, 95% 0.79 – 1.37, N= 1 RCT, 316 patients) (47), Bortezomib (HR 1.17, 95% CI 0.84 – 1.63, N= 1 RCT, 163 patients) (37), and Valproate (HR 0.85, 95% CI 0.57 – 1.27, N= 1 RCT arm) (46) has little or no effect in the overall survival of participants compared to DEC monotherapy. When comparing all-trans retinoic acid (HR 0.58, 95% CI 0.37 – 0.91, N= 1 RCT arm) and all-trans retinoic acid plus valproate (HR 0.62, 95% CI 0.40 – 0.96, N=1 RCT arm) against DEC monotherapy, patients treated with the combination therapy shown higher overall survival (S7 Figure) (46). However, we are uncertain about the true effect of these comparisons (S8 Table).

LDAC monotherapy vs LDAC combination. 

Overall survival. We identified four secondary agents from four RCTs (620 participants) reporting overall survival (29-31, 49). All the comparisons are low certainty of evidence. Venetoclax (HR 1.33, 95% CI 0.92 – 1.92, N= 1 RCT, 211 patients) (49), and Lintuzumab (HR 0.95, 95% CI 0.72 – 1.25, N= 1 RCT, 211 patients) has little or no effect in the overall survival of participants compared to LDAC monotherapy (29). When comparing volasertib (HR 1.59, 95% CI 1.00 – 2.52, N = 1 RCT, 87 patients) (30) and glasdegib (HR 2.17, 95% CI 1.44 – 3.26, N= 1 RCT, 111 patients) (31), patients treated with combination therapy shown higher overall survival (S9 Figure). However, we are uncertain about the true effect of these comparisons (S8 Table).

Complete remission. We identified four secondary agents from four RCTs (843 patients) reporting the 12-month relapse-free survival. All the comparisons are low certainty of evidence. Gemtuzumab ozogamicin plus LDAC against LDAC monotherapy (HR 1.11, 95% CI 0.73 – 1.69, N= 1 RCT, 494 participants) has little or no effect in the 12-month relapse-free survival (27). When comparing LDAC plus arsenic trioxide against LDAC monotherapy (HR 2.95, 95% CI 1.21 – 7.19, N=1 RCT, 34 participants), we found an improve of the 12-month relapse-free survival on patients treated with LDAC monotherapy (26), and when comparing vosaroxin plus LDAC (HR 0.41, 95% CI 0.16 – 1.02, N= 1 RCT, 104 participants) (28) and venetoclax plus LDAC (HR 0.58, 95% CI 0.42 - 0.80, N = 1 RCT, 211 patients) (49) we found an improvement of the 12-month relapse-free survival on patients treated with LDAC combination therapy (S10 Figure). However, we are uncertain about the true effect of these comparisons (S11 Table).

we added a discussion from one comparison without enough information to performed a robust sub-group analysis.

Treating older AML patients can be challenging, as clinicians and patients must balance the goal of increasing longevity with the risk that more aggressive treatment may increase adverse events and hospitalization. During the recommendation formulation process, with the evidence available at that time, the guideline panel found no compelling evidence of additional benefit with more aggressive treatment with more than one agent, and instances in which such therapy did increase adverse events. After the meeting, however, some new studies (RCTs and NRS) reported benefits of combinations over monotherapy, for example, DEC combined with ATRA and VPA+ATRA combination may result in better survival than DEC monotherapy [Lubbert 2020] AZA combined with venetoclax may also result in better than AZA monotherapy [DiNardo 2020]. Because these results were inconsistent with the previously identified studies, when including these new studies in the meta-analyses, the certainty of the overall evidence decreased. It is important to notice, however, that the certainty of evidence for each of these specific comparisons is low.

b) The statistics part is well written .no mention of relapse and death rate is mentioned

Thanks for bringing this situation to our attention. We made the following changes to include this information. We did not include the subgroup analysis in this section, as it was reported previously. 

- we included the following information. 

All-cause of mortality at 1 year. 

Seven RCTs (1,511 patients) addressing three comparisons reported all-cause mortality as the proportion of patient who died at 1 year (AZAM monotherapy vs LDAC monotherapy (24), LDAC monotherapy vs LDAC combination (26-30), and LDAC monotherapy vs HMAs (42, 45)). Two of the comparisons reported a reduction on mortality (AZA monotherapy vs LDAC monotherapy; [RR 0.78, 95% CI 0.64 – 0.94, N = 1 RCT, 312 patients] (24), and, LDAC monotherapy vs HMAs [RR 0.46, 95% CI 0.36 – 0.59, N = 2 NRS, 485 patients, I2 0%] (42, 45)). However, the certainty of the evidence was low, and very low, respectively, which means that we are not certain about the true effect of the interventions (S6 Table).

All-cause of mortality at 30 days. 

Seven RCTs (1,334 patients), addressing two comparisons reported all-cause mortality as the proportion of patient who died at 30 days (DEC monotherapy vs DEC plus bortezomib (37), and, LDAC monotherapy vs LDAC combination (26-28, 30, 31, 49)). The comparisons suggested have little or no difference on patient mortality at 30 days. However, the certainty of the evidence was low (S6 Table).

In page 35, we included the following information. 

Complete remission over the longest follow-up. 

5 RCTs (1,331 patients) and 1 NRS (114 patients)] addressing three comparisons reported complete remission as event-free survival (AZA monotherapy vs AZA combination (48), LDAC monotherapy vs LDAC combination (26-28, 49), and, AZA monotherapy vs DEC monotherapy (39)). One of these is moderate certainty of evidence. When comparing AZA monotherapy vs AZA combination, patients treated with AZA monotherapy shown a decrease in the event-free survival (HR 1.59, 95% CI 1.26 – 2.00, N= 1 RCTs, 488 patients) (48). One comparison is very low certainty (AZA monotherapy vs DEC monotherapy (37)) (S6 Table). There was important inconsistency in one comparison (LDAC monotherapy vs LDAC combination) (26-28, 49), which we conducted subgroup analyses (Subgroup analysis section).

c) Targeted agents were not taken into consideration

We agree with the suggestion. We consider that with the previous changes we made, the suggestion has been addressed.

---

## [Decision Letter · Decision Letter 1]

30 Nov 2021

PONE-D-21-13321R1Less intensive anti-leukemic therapies (monotherapy and/or combination) for older adults with acute myeloid leukemia who are not candidates for intensive anti-leukemic therapy: a systematic review and meta-analysis.PLOS ONE

Dear Dr. Colunga-Lozano,

Thank you for submitting your manuscript to PLOS ONE. After careful consideration, we feel that it has merit but does not fully meet PLOS ONE’s publication criteria as it currently stands. Therefore, we invite you to submit a revised version of the manuscript that addresses the points raised during the review process.Please submit your revised manuscript by Jan 14 2022 11:59PM. If you will need more time than this to complete your revisions, please reply to this message or contact the journal office at plosone@plos.org. Please include the following items when submitting your revised manuscript:A rebuttal letter that responds to each point raised by the academic editor and reviewer(s). You should upload this letter as a separate file labeled 'Response to Reviewers'.A marked-up copy of your manuscript that highlights changes made to the original version. You should upload this as a separate file labeled 'Revised Manuscript with Track Changes'.An unmarked version of your revised paper without tracked changes. You should upload this as a separate file labeled 'Manuscript'.If applicable, we recommend that you deposit your laboratory protocols in protocols.io to enhance the reproducibility of your results. Protocols.io assigns your protocol its own identifier (DOI) so that it can be cited independently in the future. For instructions see: https://journals.plos.org/plosone/s/submission-guidelines#loc-laboratory-protocols. Additionally, PLOS ONE offers an option for publishing peer-reviewed Lab Protocol articles, which describe protocols hosted on protocols.io. Read more information on sharing protocols at https://plos.org/protocols?utm_medium=editorial-email&utm_source=authorletters&utm_campaign=protocols.

We look forward to receiving your revised manuscript.

Kind regards,

Mona Pathak, PhD

Academic Editor

PLOS ONE

Journal Requirements:

Additional Editor Comments:

Thank you for revision and addressing the raised concerns, the manuscript is more clear in contents after the revisions. Still, it will be great if authors could provide an insight on the reported differential effectiveness of less intensive anti-leukemic therapies.

Reviewers' comments:

Reviewer's Responses to Questions

**Comments to the Author**

1. If the authors have adequately addressed your comments raised in a previous round of review and you feel that this manuscript is now acceptable for publication, you may indicate that here to bypass the “Comments to the Author” section, enter your conflict of interest statement in the “Confidential to Editor” section, and submit your "Accept" recommendation.

Reviewer #3: (No Response)

2. Is the manuscript technically sound, and do the data support the conclusions?

Reviewer #3: Yes

3. Has the statistical analysis been performed appropriately and rigorously? 

Reviewer #3: Yes

4. Have the authors made all data underlying the findings in their manuscript fully available?

Reviewer #3: Yes

5. Is the manuscript presented in an intelligible fashion and written in standard English?

Reviewer #3: Yes

6. Review Comments to the Author

Reviewer #3: In this article, Colunga-Lozano et al. perform a thorough and systematic retrospective review that aims to compare the efficacy and safety of low-intensity antileukemic therapy in newly diagnosed acute myeloid leukemia (AML) patients aged 55 years and above that are not candidates for intensive therapy. Although the results obtained in this study do not provide additional insights to the current ASH guidelines, the effort that the authors have made to compile, analyze, and summarize the available data is remarkable and worth to be acknowledged.

However, the authors should consider addressing possible sex-dependent effects in their analysis, which might unveil previously overlooked results in the analyzed data. AML incidence is higher in males than in females, with an age-dependent progressive increase in this difference (Cartwright et al, 2002, doi: 10.1046/j.1365-2141.2002.03750.x). Moreover, male patients show significantly worse outcomes compared to female patients (Hossain et al, 2015, doi: 10.1016/j.canep.2015.10.020) and a sex bias has been also identified in mutational profiles of AML (Juliusson et al, 2020, doi: 10.1182/bloodadvances.2019001335; Metzeler et al, 2016, doi: 10.1182/blood-2016-01-693879; Loghavi et al, 2014, doi: 10.1186/s13045-014-0074-4). Although these evidences illustrate the relevance of gender perspective in the analysis of AML clinical data, sex-specific considerations are currently not made when addressing AML management in the clinic. In this context, the omission of gender-disaggregated data in research studies perpetuate this gender gap in the therapeutic assessment of the patients. Thus, authors might want to provide detailed gender-disaggregated data (at least in the cases of moderate to high certainty evidence), which could potentially identify gender-biased differences in the efficacy and safety of the analyzed therapies and provide additional insights that could have an impact on AML patient management.

7. PLOS authors have the option to publish the peer review history of their article (what does this mean?). If published, this will include your full peer review and any attached files.

Reviewer #3: No

---

## [Author Response · Author response to Decision Letter 1]

14 Jan 2022

14th January 2022

Dr. Mona Pathak, PhD

Academic Editor

Plos One

Dear Dr. Pathak,

Thank you for taking the time to review our manuscript ID PONE-D-21-13321, and for all the thoughtful feedback. Please see below our responses (in red) to the comments regarding our manuscript, “Less intensive antileukemic therapies (monotherapy and/or combination) for older adults with acute myeloid leukemia who are not candidates for intensive antileukemic therapy: a systematic review and meta-analysis”. All changes to the manuscript are tracked.

Please let us know if anything requires further clarification.

Sincerely,

Luis Enrique Colunga Lozano

On behalf of the authors

Journal Requirements:

We reviewed our reference list. We did not identify any retracted papers and the citations are correct.

Additional Editor Comments:

Thank you for revision and addressing the raised concerns, the manuscript is more clear in contents after the revisions. Still, it will be great if authors could provide an insight on the reported differential effectiveness of less intensive anti-leukemic therapies.

Thanks for your suggestion. We added the following paragraph in our discussion in order to provide insights into the comparisons. 

Therapy selection for older adults with AML who are not candidates for intensive antileukemic therapy is based on the patient fitness, patients’ characteristics (cytogenic and molecular profiles), the trade-off between drug safety and toxicity, and patients' values and preferences. The scientific community agrees on offering therapies based on HMA agents (e.g., azacytidine, decitabine) with some exceptions: liver and kidney severe disease, prior HMA therapy, and the presence of an actionable mutation. For these populations other options are available (e.g., Low-dose cytarabine). Currently, combination therapy has become the standard of care for these patients. However, the secondary agent depends on their availability in each setting and the presence of specific genetic mutations. Venetoclax (BCL2 inhibitor) is the preferred secondary agent to add to the HMA therapies, this is based on promising results from NRS and RCTs (mentioned previously). In our review, we identify benefits from the combination therapy with venetoclax. However, the certainty of the effect was judged to be low due after creating a pooled estimate (imprecision and inconsistency). The same situation was identified with other secondary agents. We are aware that creating pooled estimates without stratifying based on the second agent may impact the effect estimate of a specific agent (e.g., venetoclax). In the comparison with enough studies, we undertook a subgroup analysis to explore their effect. However, the AZA monotherapy vs AZA combination did not have sufficient studies to explore it.

Reviewer #3: 

In this article, Colunga-Lozano et al. perform a thorough and systematic retrospective review that aims to compare the efficacy and safety of low-intensity antileukemic therapy in newly diagnosed acute myeloid leukemia (AML) patients aged 55 years and above that are not candidates for intensive therapy. Although the results obtained in this study do not provide additional insights to the current ASH guidelines, the effort that the authors have made to compile, analyze, and summarize the available data is remarkable and worth to be acknowledged.

However, the authors should consider addressing possible sex-dependent effects in their analysis, which might unveil previously overlooked results in the analyzed data. AML incidence is higher in males than in females, with an age-dependent progressive increase in this difference (Cartwright et al, 2002, doi: 10.1046/j.1365-2141.2002.03750.x). Moreover, male patients show significantly worse outcomes compared to female patients (Hossain et al, 2015, doi: 10.1016/j.canep.2015.10.020) and a sex bias has been also identified in mutational profiles of AML (Juliusson et al, 2020, doi: 10.1182/bloodadvances.2019001335; Metzeler et al, 2016, doi: 10.1182/blood-2016-01-693879; Loghavi et al, 2014, doi: 10.1186/s13045-014-0074-4). Although these evidences illustrate the relevance of gender perspective in the analysis of AML clinical data, sex-specific considerations are currently not made when addressing AML management in the clinic. In this context, the omission of gender-disaggregated data in research studies perpetuate this gender gap in the therapeutic assessment of the patients. Thus, authors might want to provide detailed gender-disaggregated data (at least in the cases of moderate to high certainty evidence), which could potentially identify gender-biased differences in the efficacy and safety of the analyzed therapies and provide additional insights that could have an impact on AML patient management.

Thanks for your suggestion. We agree with your comment, and based on it we revised our body of evidence to identify gender sub-group analyses in the comparisons with moderate certainty of the evidence. After a careful examination we concluded the following. From all the comparisons we included, we classified fifteen comparisons to have moderate certainty of evidence, from which, only two were related to efficacy outcomes (Overall survival; N=2 RCTs, Azacitidine monotherapy vs LDAC monotherapy. Complete remission; N=1 RCT, Azacitidine monotherapy vs Azacitidine combination). From the two studies included in the first comparison, only one of them reported a subgroup analysis based on gender (Dombret 2015). However, the comparisons performed by the authors did not aim the treatment group we included in our review (Azacitidine vs Conventional care regimens). The second comparisons did not perform a gender subgroup analysis for the outcome of interest (DiNardo 2020). Regarding the comparisons from the safety outcomes, neither explore a sub-group effect based on gender. We consider an important issue the suggestion raised by the reviewer. However, with our current approach will not be able to properly explore the suggestion, as the comparisons with more studies were classified as low and very low certainty of evidence.

---

## [Editor Report · Decision Letter 2]

18 Jan 2022

Less intensive anti-leukemic therapies (monotherapy and/or combination) for older adults with acute myeloid leukemia who are not candidates for intensive anti-leukemic therapy: a systematic review and meta-analysis.

PONE-D-21-13321R2

Dear Dr. Colunga-Lozano,

We’re pleased to inform you that your manuscript has been judged scientifically suitable for publication and will be formally accepted for publication once it meets all outstanding technical requirements.

Kind regards,

Mona Pathak, PhD

Academic Editor

PLOS ONE

---

## [Editor Report · Acceptance letter]

24 Jan 2022

PONE-D-21-13321R2 

Less intensive antileukemic therapies (monotherapy and/or combination) for older adults with acute myeloid leukemia who are not candidates for intensive antileukemic therapy: a systematic review and meta-analysis. 

Dear Dr. Colunga-Lozano:

I'm pleased to inform you that your manuscript has been deemed suitable for publication in PLOS ONE. Congratulations! Your manuscript is now with our production department. 

Kind regards, 

on behalf of

Dr. Mona Pathak 

Academic Editor

PLOS ONE